



# 1  Water resources management and dynamic changes in water politics
# 2  in the transboundary river basins of Central Asia

Xuanxuan Wang[1,2], Yaning Chen[1], Zhi Li[1*], Gonghuan Fang[1], Fei Wang[1,2], Haichao Hao[1,2]
[1]State Key Laboratory of Desert and Oasis Ecology, Xinjiang Institute of Ecology and Geography, Chinese Academy of
Sciences, Urumqi 830011, China
[2]University of Chinese Academy of Sciences, Beijing 100049, China
*Correspondence to*: Zhi Li (liz@ms.xjb.ac.cn)
**Abstract.** The growing water crisis in Central Asia (CA) and the complex water politics of the region's transboundary rivers
are a hot topic for research, while the dynamic changes of water politics in CA have yet to be studied in depth. Based on the
Gini coefficient, water political events and Social Network Analysis, we assess the matching degree between water and
socio-economic elements in CA and analyse the dynamics of water politics in transboundary river basins. Results indicate
that the uneven matching degree of water and land resources are the preconditions for conflicts, with the average Gini
coefficient between water and population, GDP, and cropland measuring 0.19 (completely matched), 0.47 (reasonably
matched) and 0.61 (completely mismatched). Moreover, the Gini coefficient between water and cropland increased by 0.07
over the past two decades, indicating a worsening degree. In general, a total of 591 water political events occurred in CA
with cooperation accounting for 89%. Water events have increased slightly (0.08/a) and shown three distinct stages: a stable
period (1951-1991), a rapid increase and decline (1991-2000), and a second stable period (2000-2018). Overall, water
conflicts mainly occurred in summer and winter, and the Aral Sea Basin experienced the strongest conflicts of the
transboundary river basins due to the competitive utilization of the Syr and Amu Darya rivers. The density of water
conflictive and cooperative networks in CA increased by 0.18 and 0.36 following the disintegration of the Soviet Union, and
Uzbekistan has highest degree centrality in conflictive network (6) while Kazakhstan has highest in cooperative network (15),
indicating that they have more contact with others. The findings suggest that enhancing states' cooperation and trust and
seeking support from international organizations will be helpful to eliminate conflicts and strengthen cooperation in CA.
**Keywords.** Transboundary river basins; Socio-economic development; Water politics; Social Network Analysis; Central
Asia



## 1 Introduction

With the explosive growth of the world's population and the rapid expansion of the global economy, the importance of freshwater resources is becoming increasingly obvious (Fischhendter et al., 2011; Hanasaki et al., 2013; McCracken and Wolf, 2019). There are 286 transboundary rivers in the world involving 151 countries, and both conflicts and cooperation over these water bodies are quite frequent (Zeitoun and Mirumachi, 2008; Di Baldassarre et al., 2013). Meanwhile, global warming has exacerbated the scarcity and uneven distribution of water resources, making the water-related political situation more complicated in transboundary river basins, especially in arid regions (Wolf, 1998; Zeitoun et al., 2013; Zhupankhan et al., 2017; Chen et al., 2018).

Due to many years of inappropriate transboundary water management, Central Asia (CA) is currently experiencing major contradictions between water supply and demand (Libert and Lipponen, 2012; Li et al., 2020). Most of the surface water resources originate in the mountains of the upstream countries (Tajikistan and Kyrgyzstan), while the agricultural areas are mainly located in the downstream countries (Turkmenistan, Kazakhstan and Uzbekistan), so the temporal and spatial dislocation of water and land resources has aggravated the complexity of water allocation (Rahaman and Mizanur, 2012; Wang et al., 2020a). Meanwhile, following the collapse of the Soviet Union in 1991, the hydropower allocation systems have become invalid, and political disputes were intensified due to the rise in competitive water demands for agricultural irrigation in downstream countries and hydroelectric power generation in upstream countries (Chatalova et al., 2017). Water resources have thus become the key to the security and stability of CA (Bernauer and Siegfried, 2012; Xu, 2017). The Central Asia Human Development Report by UNDP RBEC also pointed out: "the benefits from efficient use of water and energy resources could generate a regional economy twice as large and well-off 10 years from now". Moreover, researchers contend that the matching degree of water and socio-economic development is significant to water politics in CA, and that the Gini coefficient has proved to be an effective method for analyzing the matching equality between elements (Hu et al., 2016; Yu et al., 2016). This approach was also shown to be useful in analyzing the relationship between water resources and agricultural land (Hanjra et al., 2009; Liu et al., 2018), as well as the status of yield inequality (Sadras and Bongiovanni, 2004; Kisekka et al., 2017) and the rationality of land use structure (Zheng et al., 2013; Yan et al., 2016), among other issues.

The hydropolitics of transboundary rivers is emerging as a compelling research field in social hydrology (Wolf., 2007; Cabrera., 2013; Soliev et al., 2015). Some scholars have made comprehensive evaluations of water politics based on a variety of models (Wolf et al., 2004; Rai et al., 2014; Wang et al., 2015). For example, Rai et al. (2017) assessed the opportunity and risk of water-related cooperation in three major transboundary river basins in South Asia based on the fuzzy comprehensive evaluation model, while other scholars have analyzed the water politics from the perspective of historical politics (Link et al., 2016; Mollinga, 2001; Wegerich, 2008). In addition, conflictive and cooperative events are key variables for characterizing the overall state of water politics. The Transboundary Freshwater Dispute Database (TFDD) includes the water-related conflictive and cooperative events in various transboundary river basins around the world. TFDD was established by researchers at Oregon State University (Yoffe et al., 2004) and has been used for the analysis of hydropolitics



in the past few decades (Yoffe et al., 2003; Giordano et al., 2014; Gunasekara et al., 2014; McCracken and Wolf, 2019).
Based on the TFDD database, Giordano and Wolf (2002) selected three case areas – the South Asia, Middle East and
Southern Africa – to evaluate the connections between internal and external interactions of freshwater resources, and they
have found that water-related events and scales usually had different complexity and spatial variations due to specific
historical and political conditions. Eidem et al. (2012) also used the TFDD to analyze the characteristics of water politics in
the Oregon and Upper Colorado Region of the western United States, finding that cooperation was more common than
conflicts in the domestic environment. However, the TFDD database has been rarely applied in the investigation of water
politics in CA, where the water situation is critical to regional stability. Furthermore, along with the TFDD, more data
sources would be required to study the latest water political situation in CA, since most of the events recorded in the TFDD
occurred prior to 2008.
At present, related research in CA is mainly aimed at the management and allocation of water resources either sub-regionally
or across the entire region (Schlueter et al., 2013; Mazhikeyev et al., 2015; Chen et al., 2017). Sorg et al. (2014) analyzed the
impact of climate change and socio-political development on water distribution in the Syr River Basin, they suggested that
reservoirs can partially replace glaciers as water redistributors in the future. Pak et al. (2013) investigated the history of
water allocation mechanisms in the Isfara Basin and agreements on water sharing, highlighting that technical capabilities are
limiting the true implementation of agreements in the basin. Taking Uzbekistan as an example, Abdullaev and
Rakhmatullaev (2013) analyzed the transformation of water resources management in CA and concluded that the hydraulic
mission has transformed into different types of control over water management. More recently, Chang et al. (2018) looked at
the political risks of Central Asian countries based on the political risk assessment model, discovering that there are
emergent opportunities in the region as well as political risks.
However, there is yet a lack of comprehensive research about changes in the water politics of CA from the perspective of
water-related political events combined with the situation of water and socio-economic development. Therefore, in this work,
we evaluate the matching degree of water resources and socio-economic elements in CA. In so doing, we reveal the
changing process of policies and institutional structures of water management, and then further explore the dynamics of
water politics in CA's transboundary river basins by using the Social Network Analysis. Our research can serve as a
reference for policy-makers to conduct scientific water resource management and also provide new ideas for further
cooperation within Central Asian countries and beyond.
**2 Material and methods**
**2.1 Study area and transboundary rivers in CA**
Central Asia is located in the center of Eurasia and covers a total area of $400.17 \times 10^4$ km$^2$ (Fig. 1). The CA region borders
Russia to the west and north, China to the east, and Afghanistan and Iran to the south (Wang et al., 2020a). The available





water resources in CA come mainly from transboundary inland rivers (Tab.1) originating in the upper Pamirs and Tianshan
Mountains and supplied by snowmelt and glaciers. The Amu Darya River, which has the largest annual runoff in CA
$(564.00 \times 10^8$ m$^3$), is sourced in the Pamir Plateau, passes through Uzbekistan and Turkmenistan, and enters the Aral Sea at
Uzbekistan. The Syr Darya is the longest river in CA, with a length of 3,019.00 km. It originates in the Tianshan Mountains
and passes through Tajikistan, Kazakhstan and Uzbekistan until emptying into the Aral Sea (Olli and Varis, 2014).
**2.2 Data**
Data on transboundary rivers, water consumption and water volume of reservoirs in CA were obtained from the United
Nations Economic Commission for Europe (http://www.unece.org/env/water/), the United Nations Statistics Division
(https://unstats.un.org/unsd/envstats/qindicators.cshtml), the Food and Agriculture Organization of the United Nations
(http://www.fao.org/nr/water/aquastat/data/query/index), the United Nations Data Retrieval System (http://data.un.org/) and
the Portal of Knowledge for Water and Environmental Issues in Central Asia (http://www.cawater-info.net/). The population,
GDP, and cropland area data for the five Central Asian countries were obtained from the World Bank
(https://data.worldbank.org/country). Relevant data on water political events from 1951 to 2008 were obtained from the
Transboundary Freshwater Dispute Database (https://transboundarywaters.science.oregonstate.edu/), while data on water
political events from 2009 to 2018 were mainly obtained from the World Water Conflict Chronology
(https://www.worldwater.org/water-conflict/) and the Interstate Commission for Water Coordination of Central Asia
(http://www.icwc-aral.uz/events.htm).
The aforementioned Transboundary Freshwater Dispute Database (TFDD) contains data on water conflictive and
cooperative events between two or more countries in transboundary river basins around the world. The TFDD contains a
total of 6,790 events and divides them into 15 risk scales, distributed between -7 and 7. Positive values represent cooperation,
negative values represent conflict, and 0 signifies neutrality. The themes of the water events are also classified in the
database (Eidem et al., 2012). The intensity level and classification criteria of these events are shown in Fig. 2.
**2.3 Methods**
**2.3.1 Gini coefficient**
The Gini coefficient is an economic index proposed by Italian economist Corrado Gini to determine the fairness of income
distribution (Shlomo, 1979). In this study, we employ it to evaluate the matching degree of water resources and socio-
economic elements in CA. The value of the Gini coefficient ranges between 0 and 1; the closer it is to 0, the more balanced
the distribution, while the closer it is to 1, the more unbalanced the distribution. In general, 0.4 is the internationally
recognized "warning line" for the distribution gap (Dai et al., 2018). The Gini coefficient may be calculated as follows:
$G = 1 - \sum_{i=1}^{n} (x_i - x_{i-1})(y_i + y_{i-1})$ ⠀⠀⠀⠀⠀⠀⠀⠀⠀⠀⠀⠀⠀⠀⠀⠀⠀⠀⠀⠀(1)





where $G$ represents the Gini coefficient, $x_i$ represents the cumulative percentage of water consumption in the $i$-th country,
and $y_i$ represents the cumulative percentage of each economic development element, such that when $i=1$, $(x_{i-1}, y_{i-1})$ is
regarded as $(0, 0)$. The United Nations criteria for dividing the Gini coefficient are shown in Tab. 2.

**2.3.2 Matching coefficient of water and land resources**

The matching coefficient of water and land resources is defined as the amount of available water resources to cultivated land
per unit area. The larger the value of the coefficient, the better the matching degree between the distribution of water and
cultivated land resources (Zhang et al., 2018; Liu et al., 2006). The coefficient in the five CA nations is calculated following
Eq. (2):
$M_{it} = Q_{it} \times \alpha_i / S_{it}$ (2)
where $M_{it}$ is the matching coefficient of water and land resources in the $i$-th country for year $t$, $Q_{it}$ is the amount of available
water resources in the $i$-th country for year $t$, $\alpha_i$ is the percentage of agricultural water consumption in the $i$-th country, and $S_{it}$
is the arable land area in the $i$-th country for year $t$ (Liu et al., 2018).

**2.3.3 Social Network Analysis**

Social Network Analysis (SNA) is an effective method for describing the morphology, characteristics, and structure of a
network (Yuan et al., 2018). It uses graph theory and algebraic models to express various relational patterns and analyze the
impact of these patterns on the members of a network or on the whole structure of networks. It has been widely applied in
sociology, geography, information science and other fields (Hoppe and Reinelt, 2010; Tsekeris and Geroliminis, 2013). The
present research uses SNA to study the characteristics of water-related conflictive and cooperative networks, with network
density and degree centrality being common metrics in the method.
Network density analyzes the degree of connection between each node, its value is between 0 and 1, and the greater the
number of contacts, the greater the network density value. Network density is calculated following Eq. (3):
$D = \dfrac{\sum_{i=1}^{k} \sum_{j=1}^{k} d(n_i, n_j)}{k(k-1)}$ (3)
where $D$ is the network density, $k$ is the number of nodes (here, the number of countries), and $d(n_i, n_j)$ is the relational
quantity between node $n_i$ and $n_j$.
Degree centrality measures the degree of the nodes at the center of the network. In this relation, the higher the value of
degree centrality, the stronger the ability of the node to communicate with other nodes directly, and the more significant its
position in the network. The degree centrality is calculated following Eq. (4):
$C_D(n_i) = \sum_{j=1}^{n} X_{ji}$ (4)





where $C_D(n_i)$ denotes the degree centrality and $X_{ji}$ represents the connection between node $n_i$ and $n_j$. If a connection exists
between two nodes, the value is 1; otherwise, the value is 0 (Jin et al., 2010).
**3 Results**
**3.1 Matching degree between water resources and socio-economic elements in CA**
**3.1.1 Changes in the inflow and outflow of large storage facilities in CA**
Central Asia is one of the oldest irrigated areas in the world. In the modern age, numerous reservoirs and dams were built in
CA for irrigation purposes during and after the Soviet era. As a result, the natural runoff process of rivers has been disturbed
by humans and the flow pattern has changed dramatically (Karthe et al., 2015). More than 290 reservoirs with a total storage
capacity of 163.19 km$^3$ have been built in CA. In addition to irrigation, hydropower by dams accounts for up to 98% and 91%
of total electricity supplies in Tajikistan and Kyrgyzstan, respectively (Zhupankhan et al., 2017). In general, the downstream
countries have pursued irrigation independence, while the upstream countries have pursued energy independence.
In the Syr Darya River Basin, the five most significant reservoirs are the Toktogur, the Andijan, the Charvak, the Karakum,
and the Shardarya. Of these, the Toktogur, Andijan and Charvak reservoirs are located in the upstream region and the other
two reservoirs are situated downstream. The Toktogur reservoir is the largest reservoir in the Aral Sea Basin, with an
average inflow of 14.16 km$^3$/a and a release of 13.24 km$^3$/a (Fig. 3), and the flow of the Naryn River is controlled by it. The
change in the amount of released water in the Toktogur reservoir has remained relatively stable over the years. However, the
inflow first decreased and then increased as the water entering the reservoir was greatly affected by snowmelt and
precipitation in the upstream. The Andijan reservoir is located on the Kara Darya River in the Fergana Valley. The average
release of water (5.34 km$^3$/a) in this reservoir exceeds the inflow (4.82 km$^3$/a). Since the Fergana Valley is an important
agricultural region in CA, a lot of water released from reservoirs is consumed for crop irrigation. The average inflow (7.53
km$^3$/a) of the Charvak reservoir is greater than that released (7.11 km$^3$/a), with both the inflow and outflow of water showing
an increasing trend from 2010 to 2017.
The Karakum and Shardarya reservoirs are located in the lower reaches of the Syr Darya River. For these reservoirs, water
volume is greatly influenced by the upstream reservoirs. Moreover, there is a higher volume of water entering and releasing
in these two reservoirs compared to the upstream ones. The average inflow of the Karakum reservoir is 20.89 km$^3$/a and the
released water is 20.33 km$^3$/a. The average inflow of the Shardarya reservoir is 19.03 km$^3$/a, while the released water is
18.75 km$^3$/a. Both the inflow and the outflow decreased slightly from 2010 to 2017.
In the Amu Darya River Basin, the Nurek and Tuyuan reservoirs provide the main storage facilities and are located in the
upper and middle reaches of the basin, respectively. The Nurek reservoir is the second largest reservoir of the Aral Sea Basin.
It was completed in 1979 and is situated on the Vakhsh River. The average inflow (21.07 km$^3$/a) of the Nurek reservoir is





much greater than the released water (20.64 km³/a), and both the inflow and outflow exhibit an increasing trend from 2009 to
2018. For the Tuyuan reservoir, the inflow and outflow are extremely high. Furthermore, similar to the Nurek reservoir, the
average inflow of the Tuyuan Reservoir (27.35 km³/a) exceeds the amount of water being released (25.34 km³/a), with both
inflow and outflow displaying an increasing trend during 2009-2018.
At present, most of the dams and reservoirs in CA are aging and lack adequate maintenance, and there are insufficient funds
to maintain normal operation. This situation, coupled with the increasing population in the floodplain downstream,
significantly increases the water resources risk of the area. One outcome of this risk is the 2010 flooding in Kazakhstan,
which was caused by the collapse of the Kyzyl-Agash dam (Libert and Lipponen, 2012).

### 3.1.2 Spatiotemporal matching of water resources and socio-economic elements

The matching degree between water resources and socio-economic elements in CA is quite diverse. As shown in Fig. 4, the
matching degree between water resources and population was better, with an average Gini coefficient of 0.19, which was
below the warning line. However, the matching degree deteriorated from "completely matched" to "relatively matched"
during 1997-2016, with a significant increase in the Gini coefficient (surpassing the significance level of 0.05). The average
Gini coefficient between water resources and GDP was 0.47, and it also increased significantly ($p<0.05$) from 1997 to 2016,
indicating the matching degree was worsening on the whole. Specifically, the degree deteriorated first from "reasonably
matched" to "relatively mismatched" during 1997-2006, then improved from "relatively mismatched" to "reasonably
matched" during 2006-2016. These changes were mainly attributable to the Central Asian countries experiencing a great
recession in the 1990s, rendering the socio-economic situation quite severe. At present, most of the Central Asian countries
have not achieved economic transformation successfully, which has caused immense instability across most of CA
(Falkingham, 2005). The matching degree between water resources and cropland was the worst, with an average Gini
coefficient of 0.61. This not only exceeded the warning threshold but placed this element in the "completely mismatched"
category. The matching degree continued to deteriorate during 1997-2016, with the coefficient increasing from 0.56 to 0.63.
This indicates that the distribution of water and land resources in CA were imbalanced, the overall water use efficiency of
the irrigation processes was low, and that ineffective water consumption was prominent.
To further explore the matches between water and land resources, we obtained the spatial matching changes of available
water resources and cropland areas for the five CA nations based on the matching coefficient of water and land resources
(Fig. 5). Our findings indicate a large gap between the downstream and upstream countries, with the matching status of
upstream nations faring better than that of the downstream ones. Tajikistan fared the best, showing an average matching
coefficient of 2.61, followed by Kyrgyzstan with 1.96. The matching coefficients of the downstream countries were 1.30 for
Turkmenistan, 1.02 for Uzbekistan, and 0.29 for Kazakhstan. Compared with 1997, Turkmenistan's matching degree had
significantly deteriorated by 2016. However, the status of the other four countries had risen, with Kyrgyzstan showing the
highest improvement (coefficient increase of 0.52). Therefore, from these matching degrees, we can see that the quantity of





water resources was not the causation of water contradictions in CA. Rather, the issues stemmed from the uneven allocation
and utilization of water resources among these five countries.

**3.2 Changes in policies and institutional structures of water management in CA**

The former Soviet Union carried out large-scale land reclamation to increase agricultural production in CA, with water
resources being managed by the central government in Moscow. The government established the principle of division of
labor and implemented water quotas and compensation systems for losses, with the main goal of achieving maximum
economic output (Dinar, 2012). Kyrgyzstan and Tajikistan, located in the mountainous regions of the upper reaches of the
regional rivers, had abundant water resources and favorable terrain suitable for building reservoirs and developing
hydropower energy. Accordingly, those two countries undertook the task of supplying water and power to Uzbekistan,
Turkmenistan and Kazakhstan at the rivers' middle and lower reaches. Meanwhile, the downstream countries had abundant
light-heat resources, which were suitable for large-scale irrigated agriculture, and so provided agricultural, industrial and
energy products to Kyrgyzstan and Tajikistan (Micklin, 1988; Qadir et al., 2009). Thus, the upstream and downstream
countries maintained a balance of interests under the joint management of the central government.
After the collapse of the Soviet Union in 1991, the political structure of CA underwent immense changes, and the five newly
independent nations mostly disagreed on the allocation of the transboundary rivers (Kai et al., 2015). Therefore, the five
countries signed a series of contracts and established new institutions for the reallocation and management of water
resources (Zhi et al., 2015). In February 1992, the Interstate Commission on Water Coordination (ICWC) was established in
"agreement on cooperation in joint management, use and protection of water resources of inter-state sources", which was
responsible for determining the water releasing mechanism of reservoirs and allocation of water resources in the Amu and
Syr Darya river basins. In 1993, CA established the International Fund for Saving the Aral Sea (IFAS) to meet
environmental and ecological challenges in the Aral Sea Basin and realize sustainable development. In addition, the Inter-
State Commission on Sustainable Development (ICSD) was established as an "agreement on joint action to address the
problem of the Aral Sea and surrounding areas, environmental improvement and ensuring socio-economic development of
the Aral Sea region" in 1993. The ICSD essentially managed the socio-economic activities and ecological environment of
the Aral Sea Basin. Then, during the reorganization of the institutions in 1997, both the ICWC and ICSD became a part of
the IFAS. The evolution of water management structures in CA can be seen in Fig. 6.
For domestic water management, the Central Asian nations established specialized agencies. Kyrgyzstan created the
Ministry of Emergency Services, and Tajikistan followed the model of water resources management in Kyrgyzstan and
established the Ministry of Energy and Water Resources. However, Tajikistan and Kyrgyzstan are the two poorest countries
in CA, and because of their economic shortfalls, many policies were difficult to implement. In addition, the water policies of
these two countries have always been linked to poverty reduction and economic benefits, so their focus was different from
that of the other three countries (Yuldashev and Sahin, 2016).



In the formulation of water policies, Kazakhstan has continuously assigned the authority of water management to the
Ministry of Agriculture, the Ministry of Environmental Protection, and the Ministry of Energy in different periods. In 2019,
Kazakhstan established the Ministry of Ecology, Geology and Natural Resources. Meanwhile, Uzbekistan and Turkmenistan
had both previously established Ministries of Agriculture and Water Resources, but the management of water and agriculture
has now been separated. Specifically, Turkmenistan established the Ministry of Water Resources and Uzbekistan the
Ministry of Emergency Services to manage water. In terms of water fees, Turkmenistan has implemented a free water policy,
while the other four countries have founded the Water Users Association (WUA) to provide financial subsidies for irrigation
water.
**3.3 Dynamics of water political events in the transboundary river basins of CA**
**3.3.1 Changing trends of water conflictive and cooperative events**
From 1951 to 2018, a total of 591 water political events occurred in the transboundary river basins in CA, including 53
conflictive events, 528 cooperative events, and 10 neutral events (Fig. 7). The number of cooperative events accounted for
89.34% of the total water events, which far exceeded the number of conflictive events, indicating that cooperation was more
common than conflicts in CA. Over the past 60 years, the number of water political events has increased slightly on the
whole at an increasing rate of 0.08/a, with the changes showing three significant stages. From 1951 to 1991 (the former
Soviet Union), water events decreased slightly and the fluctuation range was stable. However, the fluctuation in water events
increased dramatically after the collapse of the Soviet Union in 1991, with the number of water events increasing rapidly and
reaching their highest value (77) in 1997. This was likely due to the countries being eager to explore the water policies that
were most suitable for the post-Soviet era, and because of this exploration of policies, cooperation among the countries was
occasionally marred by conflicts over the short term. Then the number of water events has declined rapidly after 1997. From
2000 onward, the changes gradually stabilized.
**3.3.2 Spatial variations in water conflictive and cooperative events**
There were obvious differences in the water political events in various transboundary river basins in Central Asia (Fig. 8).
As a hot spot in water politics, the Aral Sea Basin had the largest number of water events (227), accounting for 71.16% of
total events in CA. The basin was also the site of the most water conflicts (24 events). The major water issues in the basin
included the distribution and management of water resources in the Syr and Amu Darya rivers and the construction of large
reservoirs. During the same time frame, there were 18 water-related events in the Ob River Basin shared by Kazakhstan,
Russia and China. The main themes underlying these events were water quantity and hydropower. Elsewhere in CA, in the
region where the Ili River rises from the Khan Tengri Peak of the Tianshan Mountains and crosses China and Kazakhstan
before flowing into Balkhash Lake, 13 water political events occurred, 12 of were cooperative. The main themes of these
events were water distribution and navigation. As well, there were 10 water events in the Tarim River Basin (all cooperative),





with water quantity as the theme. Finally, where the Ural River flows through Russia and Kazakhstan to the Caspian Sea,
only 3 water events were recorded.

### 3.3.3 Network building of water conflictive and cooperative events between CA and other countries

Before the collapse of the Soviet Union (1951-1991), the water conflictive network spread across neighboring countries,
with the Soviet Union as the core. The network covered Europe, Asia, Africa, South America and North America (Fig. 9a) at
a density of 0.20 (Tab. 3). The countries that had the most water conflicts with the Soviet Union were Egypt (6), followed by
the United States and China (5). However, few conflicts erupted between Kyrgyzstan, Tajikistan and Uzbekistan within the
Soviet Union. The disintegration of the Soviet Union had a great impact on the water political structure in CA, and the water
conflictive network has since been distributed in a crisscross pattern (1992-2018), with the five Central Asian countries as
the core (Fig. 9b). Moreover, the network density has increased to 0.38, indicating that the conflictive intensity has increased.
In terms of the degree centrality (Tab. 4), Uzbekistan is at the core of the water conflictive network, with a centrality of 6,
followed by Kazakhstan and Tajikistan with degree centralities of 5 and 4, respectively. Water conflicts between Kyrgyzstan
and Uzbekistan have been most frequent (9 conflictive events). This is mainly because Kyrgyzstan borders Uzbekistan and
shares the Syr and Amu Darya rivers, giving cause for greater competition of water resources. Furthermore, the matching
degree of land and water resources in the two countries is quite different, which in itself foments conflicts.
The number of conflictive events between Kyrgyzstan and Tajikistan numbered 7, between Kazakhstan and Kyrgyzstan
numbered 6, while Tajikistan and Turkmenistan experienced 3 water-related conflictive events. The neighboring countries
that had conflicts with the Central Asian countries were mainly Russia, Azerbaijan and China, with most of the events (6)
occurring between Russia and CA (Kazakhstan and Russia 4, Tajikistan and Russia 2). Overall, there were 3 water-instigated
conflictive events between Central Asian countries and China.
The networks of water cooperation were more complex than water conflicts. Moreover, the scope of water cooperation in the
former Soviet Union was very wide, linking to 32 countries around the world (Fig. 9c) and involving six continents (Asia,
Europe, Africa, Oceania, North America and South America). Although these networks centered on the Soviet Union and
radiated outward, the network density was small (only 0.06). The largest number of cooperative events with CA was linked
to Egypt (41), followed by Iran (32) and China (22). From 1992 to 2018, the scope of water cooperation became more
concentrated; at the same time, the intensity of cooperation also greatly increased and the networks grew denser (up to 0.42)
(Fig. 9d).
Overall, Kazakhstan showed the highest degree centrality (15), indicating that it played the most prominent role in the
cooperative network and had most frequent water cooperation with other countries. Turkmenistan and Uzbekistan showed
less cooperation with others, and both had a degree centrality of 12. Cooperation was mainly distributed between the five
Central Asian countries, and water-related events among them were far more than those with other countries. Specifically,
the number of water cooperative events between Kazakhstan and Kyrgyzstan was the highest (280), followed by





Kazakhstan-Tajikistan, and Kyrgyzstan-Tajikistan, which saw 260 cooperative events each. Meanwhile, CA had cooperative
relations with 12 countries around the world, among which those with western neighbors were more intensive, such as
Russia and Ukraine. Russia has a very significant relationship with CA for historical reasons, and it is also the key trading
partner of CA (Cooley, 2009). The eastern neighboring country that cooperated the most with CA was China. Other than for
Turkmenistan, the other four Central Asian countries all had cooperative relations with China, with a total of 29 water
cooperative events being recorded.

### 314 3.3.4 Intensities and themes of water conflictive and cooperative events

Fig. 10a depicts the distribution of different degrees of water political events, the green bars indicate cooperative events
(with degrees of 1 to 7), the orange bars indicate conflictive events (with degrees of -1 to -7), and the white bar indicates
neutral events (with a degree of 0). Among cooperative events, all levels occurred except level 7, with the highest number of
water events occurring at level 4 (non-military agreement) for a total of 152 events or 28.79% of all cooperative events. This
was followed by level 1 (135), accounting for 25.57% of all events. Level 5 had the lowest number (6), accounting for just
1.14% of the total. In general, water cooperative events are dominated by weak level cooperation, with less cooperation at
deeper levels.
Among water conflictive events, however, all levels occurred except levels -7 and -6. Furthermore, level -2 water events
(strong/official verbal hostility) accounted for the highest number of all conflictive events (28.30%), at a total of 15. Level -4
water conflictive events occurred the least and only accounted for 7.55%. From these data, we can see that water conflicts in
CA were dominated by weak levels that mainly belonged to official or unofficial verbal hostilities, but that no conflicts
occurred at strong levels. These reasonably good relations between and among Central Asian countries provided a good
foundation for deeper cooperation in the future.
Water political events in CA involved a variety of themes. Among these, water quantity was the most common, accounting
for 42.00% of all conflictive events (Fig. 11a). Due to a lack of communication and trust, the allocation of water quantity in
the region's transboundary rivers was the primary cause of the water conflicts in CA, especially between upstream and
downstream countries. The second most dominant theme was infrastructure and development (26.00%), which included
infrastructure construction and development of projects, such as dams and canals. The construction of water conservancy
facilities was also quite controversial in CA because it had a direct and far-reaching effect on the available water resources in
each country, especially with regard to large reservoirs and dams. In addition, the months in which the water conflicts took
place also differed among the Central Asian nations (Fig. 10b), January experienced the highest number of water conflicts (a
total of 9), followed by July (8). Seasonally, water conflicts were most likely to occur in summer and winter, accounting for
33.96% and 26.42% of all water conflictive events, respectively. The main reason for these occurrences is that water demand
for irrigation and hydropower is highest in those two seasons, leading to more conflicts.





Different from water conflicts, joint management was the major theme in water cooperation (Fig. 11b), accounting for 31.12%
of all themes involving cooperative events. As a means to resolve disagreements and conflicts in the allocation of water
quantity, Central Asian countries formulated joint management measures for transboundary rivers. The theme of joint
management was followed by infrastructure and development (17.22%), with water quantity being the third largest theme
(14.73%). Next in line, water quality accounted for 11.62% of the events and mainly included those related to environmental
concerns. Flood control/relief (0.57%) and economic development (0.19%) accounted for lowest proportion of water
cooperative events.
**4 Discussion**
The water resources of the transboundary rivers in CA have undergone the unified distribution and negotiated management
successively, and water politics in general has changed dramatically. In our study, the water political pattern in CA was
dominated by water cooperation, with water conflictive events accounting for only 8.97% of all water-related events. This
spread was basically consistent with the overall trend in water politics in the global transboundary river basins. Wolf et al.
(2003) found that over 2/3 of global water political events were cooperative, while less than 1/3 were categorized as conflicts,
and most of conflictive events were "mild". However, we have further found that although water cooperation had clear
advantages, the levels of cooperation were mainly weak (especially among the five Central Asian countries), indicating that
the achievements in cooperation in CA are not currently obvious. Furthermore, with the factors of climate change,
population growth and degradation of water and land resources, the matching degree of water and socio-economic
development worsened, thus intensifying the competition for water resources among Central Asian countries.
In terms of water management policies, although the countries in CA have experienced several reforms and innovations, the
existing mechanisms still have some drawbacks. The first of these is that the five countries divided the management of their
water resources into different departments, and the management authority of each country was very different. Consequently,
there was no effective cooperation mechanism between the countries, resulting in low cooperative efficiency. Secondly, the
existing water policies mostly targeted surface water resources (e.g., transboundary rivers) while showing a lack of effective
unified management and planning of groundwater (Zhang et al., 2014; Fang et al., 2018). Moreover, although IFAS has been
an effective organization to save the Aral Sea, it still has many institutional weaknesses. For instance, there has been a
consistently low level of information exchange between IFAS and its subordinate organizations (ICWC and ICSD) (Janusz-
Pawletta, 2015), and the focus of policies formulated by member countries has been quite different.
Among the transboundary river basins in CA, the Aral Sea Basin has faced the most serious water crisis and most complex
water politics, so many studies thus far have focused on the water issues in the Aral Sea (Micklin, 2010; Shi et al., 2014;
Zhang et al., 2019). In fact, the dramatic retreat of lake volume and degradation of aquatic ecosystem have made the Aral
Sea a world-renowned "Ecological Disaster Area" (Wang et al., 2020b). According to our study, there have been 24 water





conflictive events in the Aral Sea Basin, accounting for 45.28% of the total conflictive events in CA. Within the basin, the Ferghana Valley, located at the border of Uzbekistan, Tajikistan and Kyrgyzstan, is prone to water conflicts due to complex ethnic issues and competition for water and arable land. For example, in 1990, an outbreak of violence over water competition in the Kyrgyzstan town of Osh on the border with Uzbekistan killed 300 people. Megoran (2004) indicated that the dispute in the Ferghana Valley facilitated the consolidation of the authoritarian regime in Uzbekistan, and also provided opportunities for anti-minority propaganda in Kyrgyzstan. Meanwhile, there have been many conflicts between upstream and downstream countries over water-energy exchanges in the Aral Sea Basin. For instance, the Parliament of Kyrgyzstan passed a law that classified water as a commodity in June 2001, and announced that downstream countries had to be charged for water from that point onward. In response, Uzbekistan cut off all deliveries of natural gas to Kyrgyzstan. In 2012, Uzbekistan also cut off natural gas deliveries to Tajikistan in response to the construction plan of the Rogun Dam in Tajikistan, which Uzbekistan said would disrupt its water supplies.

In contrast, water politics in the Ili River Basin was dominated by cooperation, with water cooperative events accounting for 92% of all water-related events. About 85% of the basin is located in Kazakhstan and about 15% in China (Zhupankhan et al., 2017). There have been 13 water events in the Ili River Basin, 8 of which have been related to China (China-Kazakhstan/China-Kyrgyzstan), and 7 of which are categorized as water cooperation. In fact, the overall level of cooperation has been relatively high in this region, focusing on the allocation of water quantity in the Ili River (Tab. 5). Meanwhile, Duan et al. (2020) demonstrated that water flowing from the upper reaches of the Ili River in China to Kazakhstan had increased from 1931 to 2013. These findings provided a positive reference for the cooperation and management of transboundary rivers in CA.

In general, to eliminate conflicts and strengthen cooperation in CA, the following approaches would be effective. First of all, the successful management of transboundary rivers in CA depends on enhancing the countries' cooperation and trust (Libert and Lipponen, 2012; Janusz-Pawletta, 2015). Although there has already been a series of agreements on joint management of water resources, all of the countries essentially aimed to maintain their own interests rather than abide by the full terms of the agreements. Therefore, we suggest that CA learn from the water sharing agreement of the Senegal River Basin in West Africa (World Water Development Report 2003). In this seminal agreement, each riparian country must notify other countries before undertaking any project or measure that could affect the water availability of adjoining countries. Such an approach would reduce many unnecessary conflicts. Moreover, in future management agreement, the countries involved should not only focus on their own interests. Instead, they should work together to maximize the total benefits of transboundary river basins, such as establishing common electricity and energy markets and addressing environmental issues jointly.

Secondly, the making of water allocation policies should think more about the effect of climate change. Climate change has brought great uncertainty to water resources and has accelerated ecological deterioration, these issues will likely exacerbate future water conflicts, so more time-sensitive water allocation models must be adopted. In addition, the countries involved





should consider making full use of the assistance of international and regional organizations (Wegerich, 2004). Relying
solely on their own strength, the five Central Asian countries may suffer the same low cooperation efficiency they have
experienced in the past. Therefore, they should actively seek financial and technical support from organizations such as the
United Nations Development Programme (UNDP), the Shanghai Cooperation Organization (SCO), the Asian Development
Bank (ADB), and others. Furthermore, CA should deepen its cooperation with neighboring countries such as China and
Russia.
**5 Conclusions**
We assessed the matching degree of water resources and socio-economic elements, and analyzed the dynamic changes of
hydropolitics in transboundary river basins in CA, the findings are as follows:
The average Gini coefficient of water resources and population was smallest (0.19), indicating a better matching degree,
while the degree deteriorated from "completely matched" to "relatively matched" during 1997-2016. The average Gini
coefficient of water resources and GDP was 0.47 and belonged to the "reasonably matched", but this coefficient increased
significantly during 1997-2016. The average Gini coefficient of water resources and cropland was the worst (0.61) and
belonged to "completely mismatched", with the degree further deteriorating during 1997-2016. Spatially, the matching
coefficients of water and land resources in Tajikistan (2.61) and Kyrgyzstan (1.96) were higher than downstream countries,
indicating a better matching status, whereas Turkmenistan worsened from 1997 to 2016. Generally, the imbalanced
distribution of water and land resources was the spark that ignited various water-related political crises in CA.
Overall, there were 591 water political events in CA, with cooperative and conflictive events accounting for 89.34% and
8.97%, respectively. During 1951-2018, the events increased slightly at a rate of 0.08/a, rising rapidly from 1991 and then
dropped dramatically from 1997 onward. The Aral Sea Basin experienced the most water events (227) of all transboundary
river basins in CA, along with the strongest conflicts (accounting for 45.28% of all conflictive events). Conflictive events
mainly occurred in summer and winter with water distribution the major issue. While joint management of transboundary
rivers was the major issue of cooperation.
In the structure of conflictive networks, the density increased by 0.16 after the disintegration of the Soviet Union in 1991.
Uzbekistan had the largest degree of centrality (6) and formed the core of the network. In cooperative networks, the density
increased from 0.06 to 0.42, with Kazakhstan showing the highest degree of centrality (15). Conflictive events between
Kyrgyzstan and Uzbekistan were most (9) while cooperative events between Kazakhstan and Kyrgyzstan were most (280).
Both water conflicts and cooperation remained mainly at weak levels, with strong/official verbal hostility (level -2) and non-
military agreement (level 4) having the largest proportion of water conflictive and cooperative events. Strengthening
cooperation and trust, considering the impact of climate change and seeking financial and technical support from
international organizations would be helpful to eliminate conflicts and promote cooperation for CA.




**Code/Data availability**

The data is available on request from the corresponding author (liz@ms.xjb.ac.cn).

**Author contribution**

Xuanxuan Wang: Conceptualization, Methodology, Software, Data curation, Writing-original draft preparation. Yaning Chen: Conceptualization, Writing-review & editing, Supervision. Zhi Li: Validation, Supervision, Writing-review & editing. Gonghuan Fang: Writing-review & editing, Supervision. Fei Wang, Haichao Hao: Writing-review & editing.

**Competing interests**

The authors declare that they have no conflict of interest.

**Acknowledgements**

The research is supported by the Strategic Priority Research Program of the Chinese Academy of Sciences (XDA19030204) and the National Natural Science Foundation of China (U1903208). The authors gratefully acknowledge the Youth Innovation Promotion Association of the Chinese Academy of Sciences (No. 2018480).

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







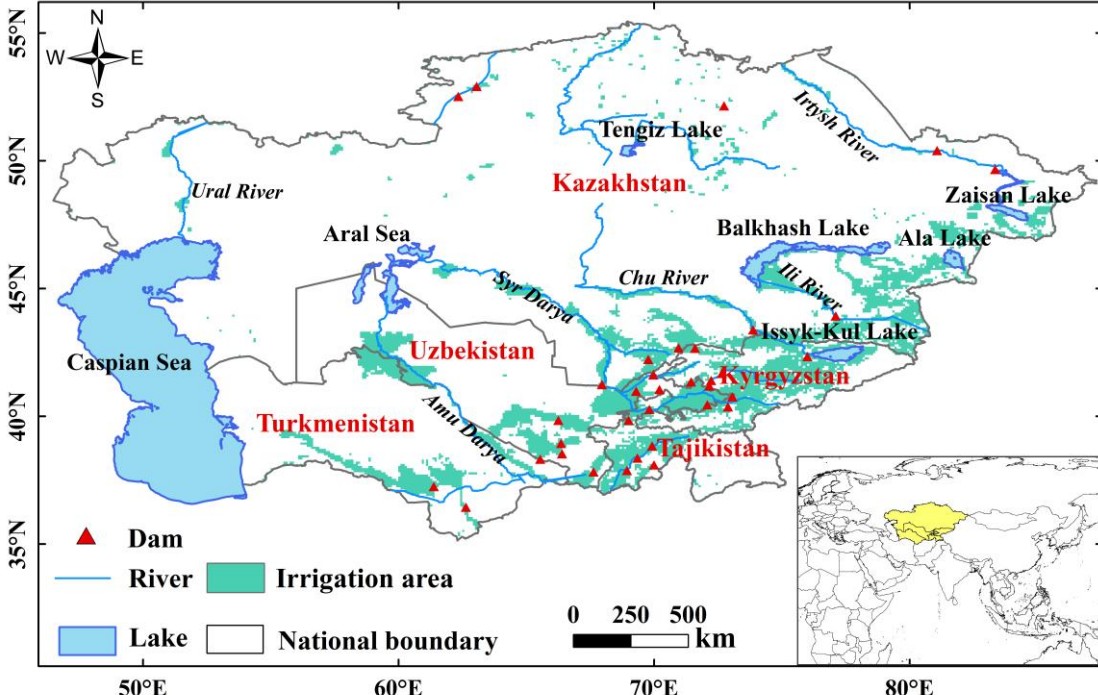


**Figure 1: Location of Central Asia. This map is made with ArcGIS, and all layers are from the public layers. The world shapefile and country borders are from the layers in ArcGIS online (https://www.igismap.com/download-world-shapefile-free-country-borders-continents/), the lake outlines are from the Natural Earth Data (http://www.naturalearthdata.com/), and the raster file of irrigation area is from the Food and Agriculture Organization of the United Nations (http://www.fao.org/aquastat/en/geospatial-information/global-maps-irrigated-areas).**







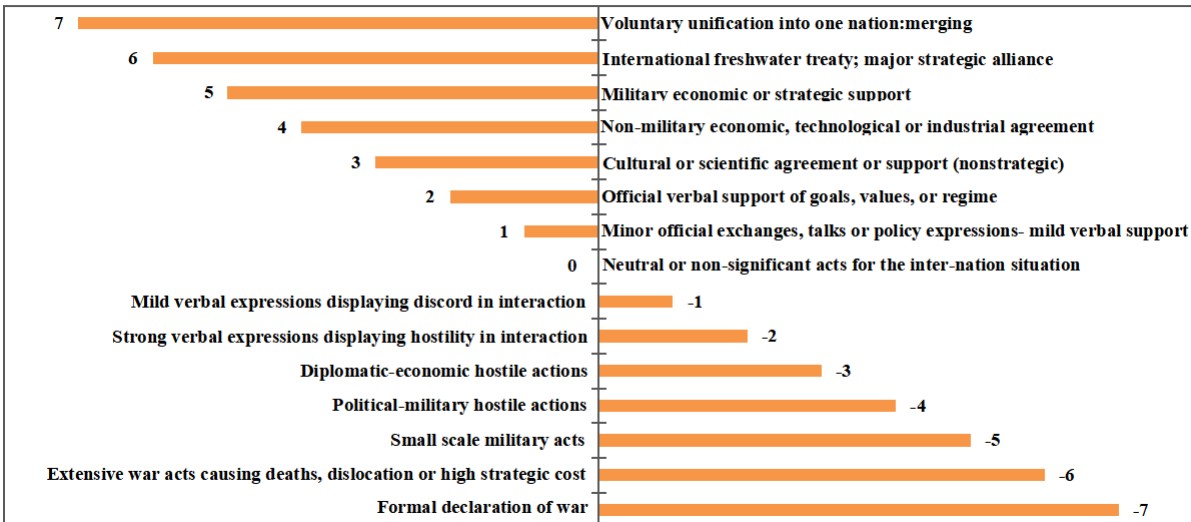

**Figure 2: Classification criteria for water-related political events.**





**Figure 3: Changing inflow and outflow trends of major reservoirs in Central Asia.**







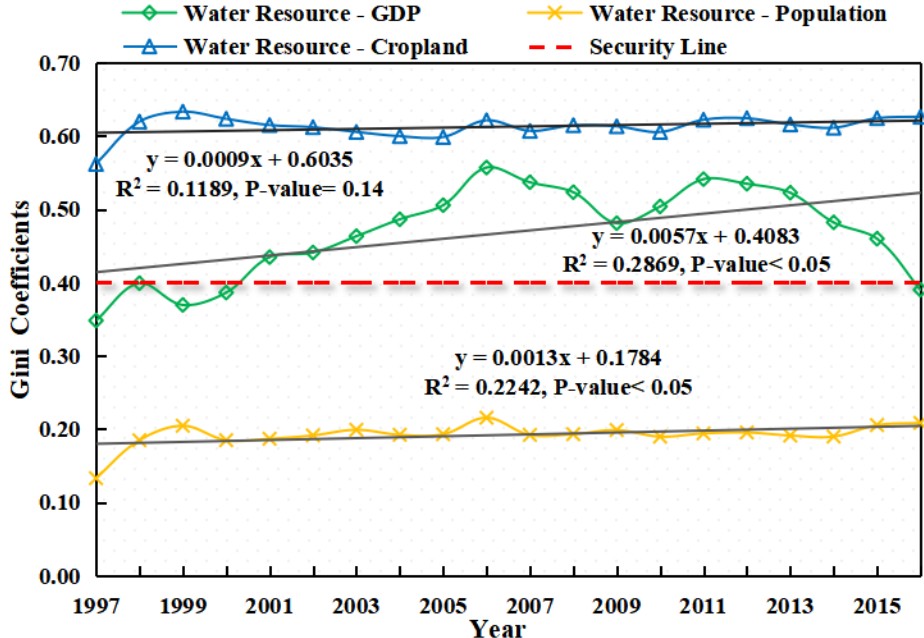


**Figure 4: Variations in Gini coefficient between water resources and socio-economic elements in Central Asia from 1997 to 2016.**







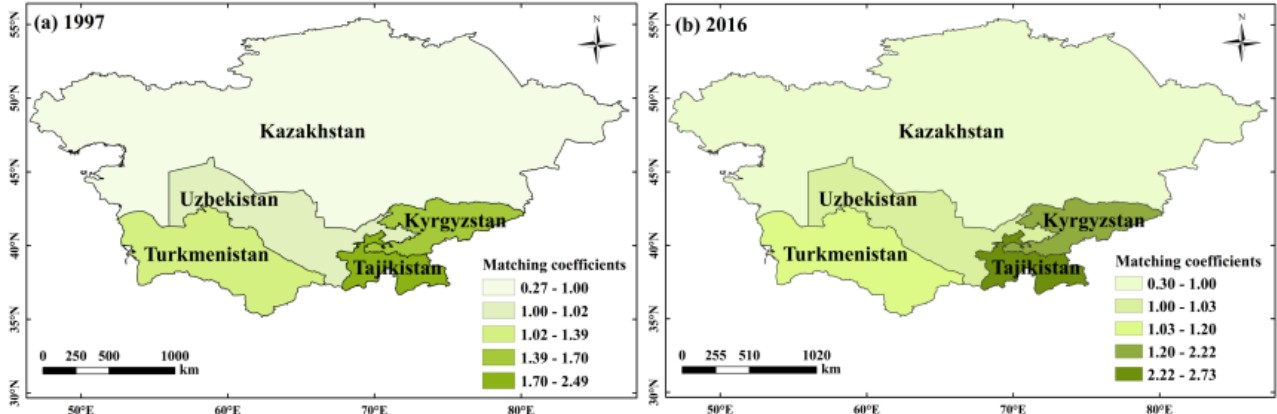


**Figure 5: Spatial distribution of matching coefficients of water and land resources in the five Central Asian countries in (a) 1997 and (b) 2016. The country borders are from the public layer in ArcGIS online (https://www.igismap.com/download-world-shapefile-free-country-borders-continents/).**





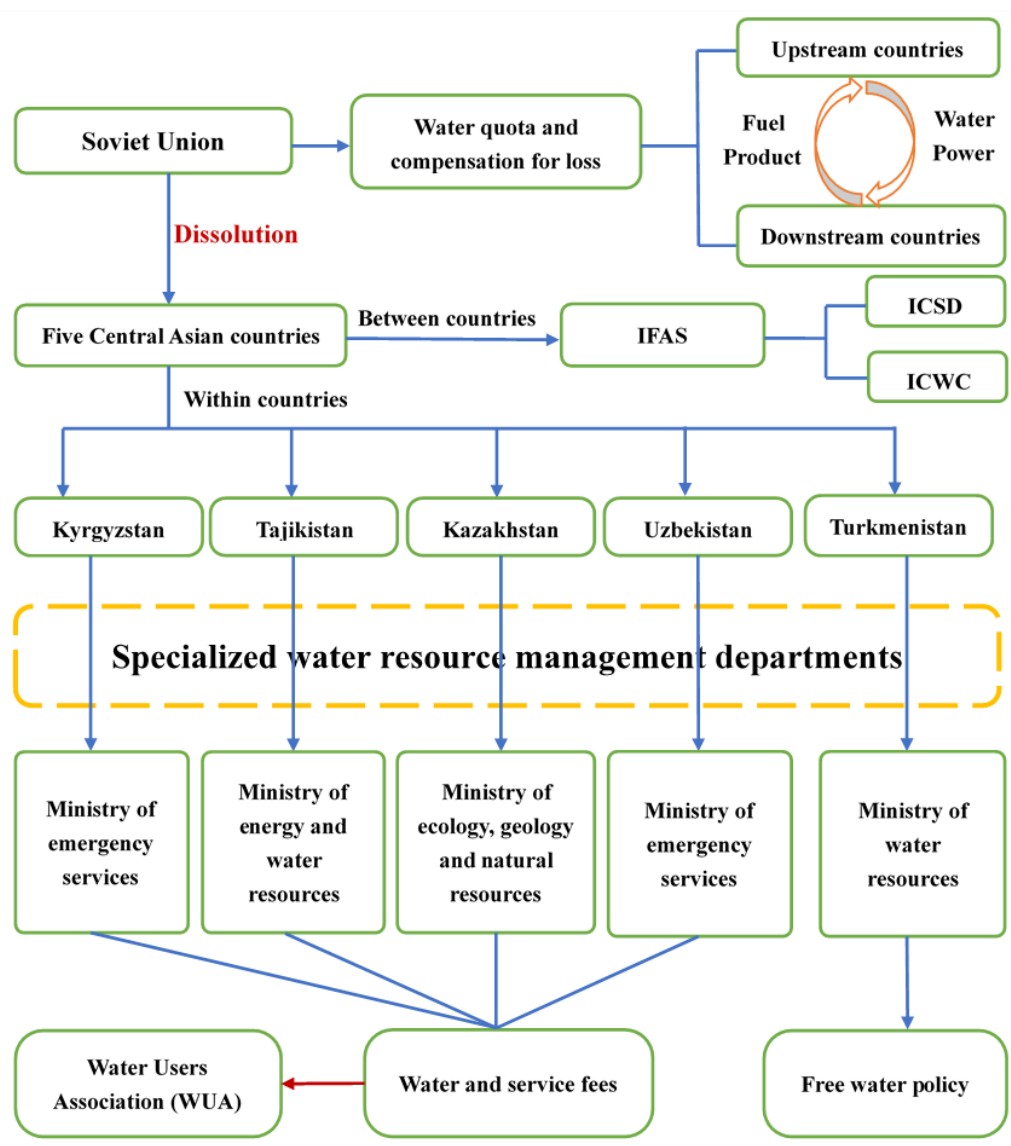


**Figure 6: Evolution of water management policies and institutional framework in Central Asia.**







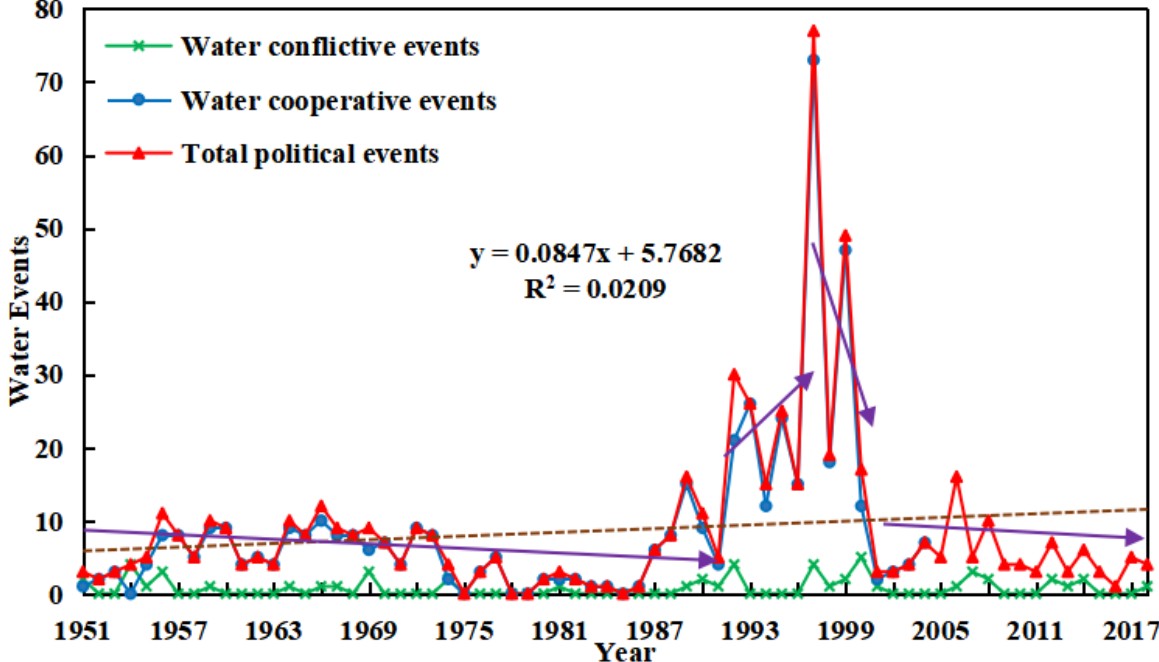


**Figure 7: Changing trends in water conflictive, cooperative and total political events in Central Asia from 1951 to 2018.**





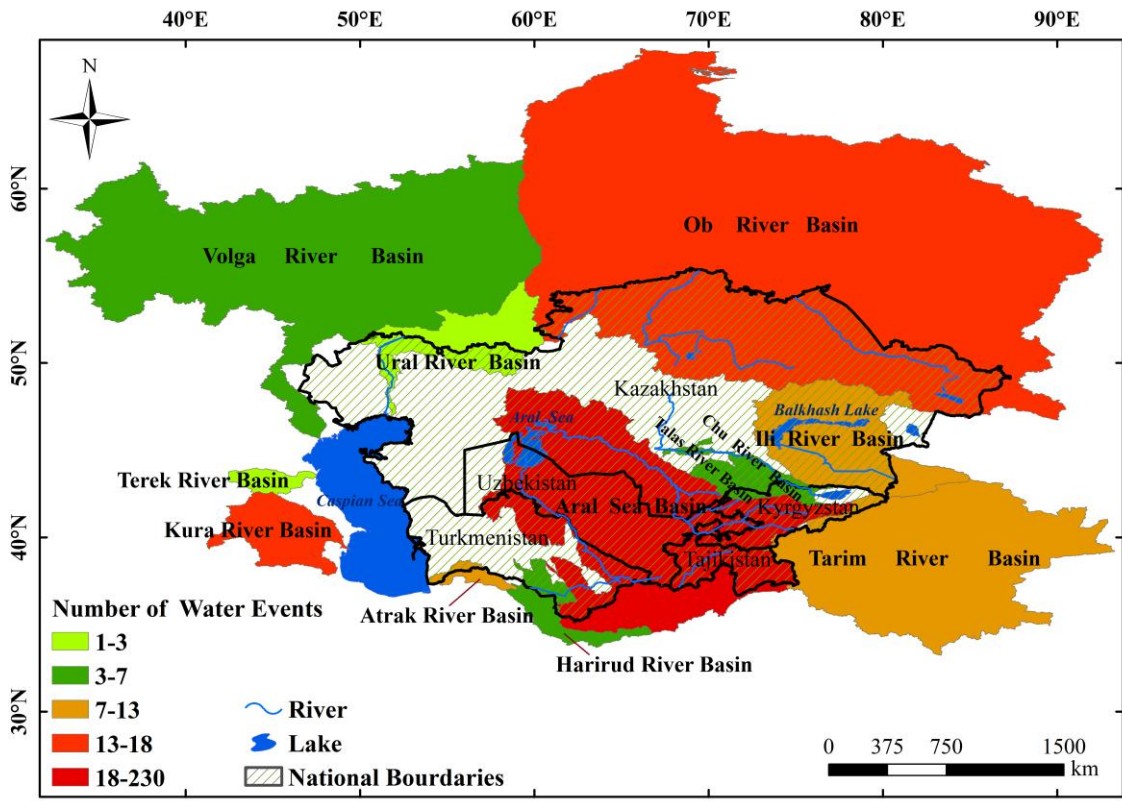


Figure 8: Spatial distribution of water political events in transboundary river basins in and around Central Asia from 1951 to 2018. The country borders are from the public layer in ArcGIS online (https://www.igismap.com/download-world-shapefile-free-country-borders-continents/), and the river basin borders are from the Transboundary Waters Assessment Programme (http://twap-rivers.org/).

638

639

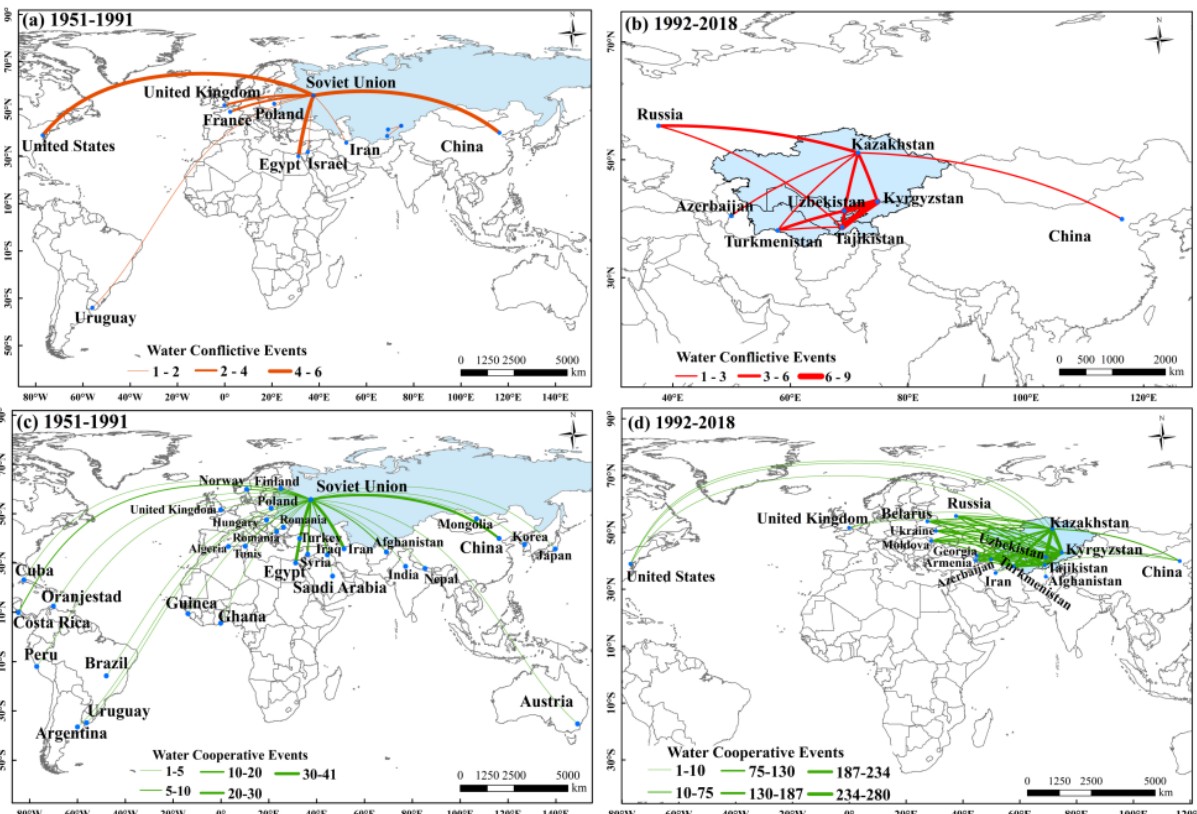

**Figure 9: Water conflictive and cooperative networks between Central Asian countries and other nations in the world: (a) Number of water conflictive events in 1951-1991 and (b) 1992-2018; (c) number of water cooperative events in 1951-1991 and (d) 1992-2018. The world shapefile and country borders are from the public layers in ArcGIS online (https://www.igismap.com/download-world-shapefile-free-country-borders-continents/).**





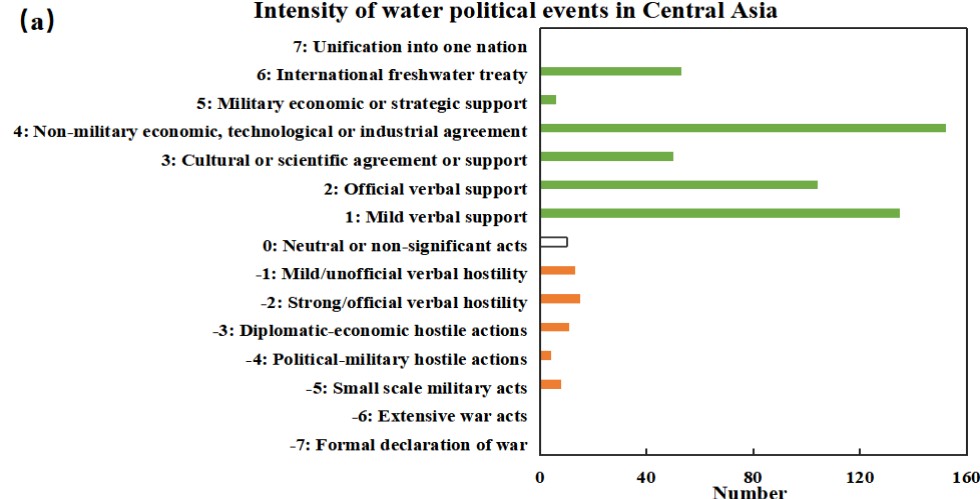

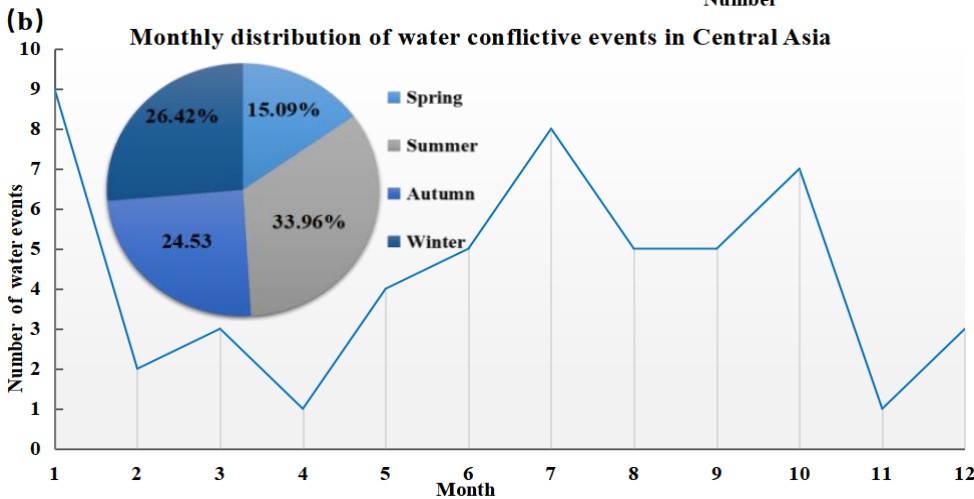


**Figure 10: Graph showing (a) number of water political events in Central Asia according to intensity and (b) monthly distribution**
**of water conflictive events.**







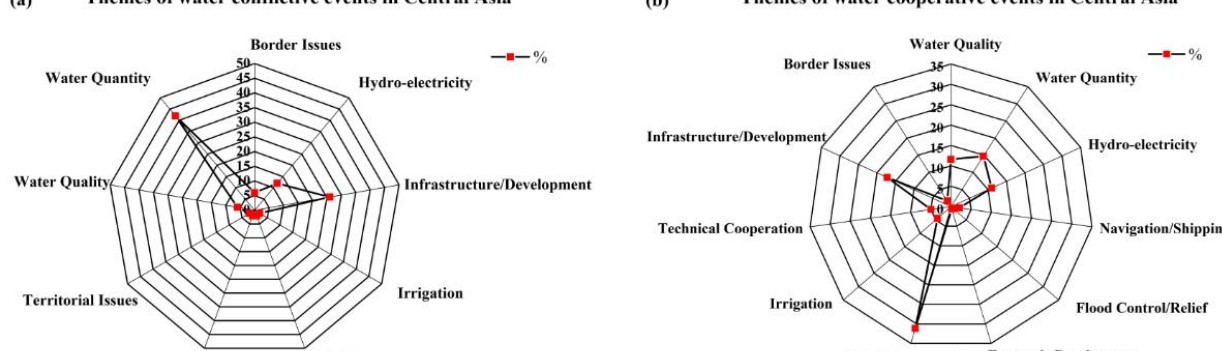


**Figure 11: Percentages of (a) water conflictive and (b) cooperative events in Central Asia according to theme.**



**Table 1: Transboundary Rivers and Tributaries in Central Asia.**

| River/tributary | Length (km) | Area of the basin ($10^4$ km²) | Average flow (m³/s) | Annual runoff ($10^8$m³) | Riparian countries | Recipient |
|---|---|---|---|---|---|---|
| **Amu Darya** | 2540.00 | 46.50 | 1970.00 | 564.00 | AFH,KGZ, TJK,UZB, TKM | Aral Sea |
| -Surkhan Darya | * | 1.35 | 74.20 | 33.24 | TJK,UZB | Amu Darya |
| -Kafirnigan | * | 1.16 | 170.00 | 54.52 | TJK,UZB | Amu Darya |
| -Pyanj | 1137.00 | 11.35 | 1012.00 | 430.00 | AFH,TJK | Amu Darya |
| -Vakhsh | 524.00 | 3.91 | 1012.00 | 202.00 | KGZ,TJK | Amu Darya |
| **Zeravshan** | 877.00 | 1.80 | 161.00 | 51.37 | TJK,UZB | Desert |
| **Syr Darya** | 3019.00 | 78.26 | 1060.00 | 341.00 | KGZ,UZB, TJK,KAZ | Aral Sea |
| -Naryn | 807.00 | 5.91 | 381.00 | 135.30 | KGZ,UZB | Syr Darya |
| -Kara Darya | 180.00 | 2.86 | 122.00 | 39.21 | KGZ,UZB | Syr Darya |
| -Chirchik | 161.00 | 1.42 | 104.00 | 79.49 | KGZ,UZB KAZ, | Syr Darya |
| -Chatkal | 217.00 | 0.71 | 115.00 | 2.71 | KGZ,UZB | Chirchik |
| **Chu** | 1186.00 | 6.25 | 130.00 | 66.40 | KGZ,KAZ | Desert |
| **Talas** | 661.00 | 5.27 | 27.40 | 18.10 | KGZ,KAZ | Desert |
| **Ili** | 1236.00 | 15.10 | 374.20 | 126.00 | CHN,KAZ | Balkhash Lake |
| **Murgab** | 978.00 | 4.69 | 50.00 | 16.57 | AFH,TKM | Desert |
| **Tejen** | 1150.00 | 7.03 | 24.00 | 7.57 | AFH,IRI,T KM | Desert |

**Note: AFH- Afghanistan, CHN- China, IRI- Iran, KAZ- Kazakhstan, TJK- Tajikistan, KGZ- Kyrgyzstan, TKM- Turkmenistan, and UZB- Uzbekistan; * means no data**

**Table 2: Division of Threshold Value of Gini Coefficient.**

| Extent | 0 | 0< $G$< 0.2 | 0.2 ≦ $G$< 0.3 | 0.3 ≦ $G$< 0.4 | 0.4 ≦ $G$< 0.5 | 0.5 ≦ $G$< 1 | 1 |
|---|---|---|---|---|---|---|---|
| **Rank** | Highly matched | Completely matched | Relatively matched | Reasonably matched | Relatively mismatched | Completely mismatched | Highly mismatched |

**Table 3: Density of Water Conflictive and Cooperative Network in Fig. 9.**

| Network | Period | Density | Standard Deviation |
|---|---|---|---|
| Conflicts | 1951-1991 | 0.20 | 0.40 |
| | 1992-2018 | 0.38 | 0.48 |
| Cooperation | 1951-1991 | 0.06 | 0.23 |
| | 1992-2018 | 0.42 | 0.49 |



662

**Table 4: Degree centrality of water conflictive and cooperative network for the five Central Asian countries after the collapse of the Soviet Union (1992-2018).**

| Water conflictive network | | Water cooperative network | |
|---|---|---|---|
| Country | Degree centrality | Country | Degree centrality |
| Uzbekistan | 6 | Kazakhstan | 15 |
| Kazakhstan | 5 | Kyrgyzstan | 14 |
| Tajikistan | 4 | Tajikistan | 14 |
| Kyrgyzstan | 3 | Turkmenistan | 12 |
| Turkmenistan | 3 | Uzbekistan | 12 |

665

**Table 5: Water-related Political Events in the Ili River Basin Between China and Central Asian Countries.**

| Date | Country List | Event Intensity | Event Type | Description |
|---|---|---|---|---|
| 1993/1/1 | CHN_KGZ | 2 | Water quantity | China broaches signatory Kyrgyzstan with possibility of exploiting 4 rivers whose waters are shared by Xinjiang in Western China and Kyrgyzstan. |
| 1993/1/1 | CHN_KAZ | 4 | Water quantity | Kazakhstan and China agree to build water conservancy works over the Horgos River. |
| 1993/1/18 | CHN_KAZ | 4 | Water quantity | China and Kazakhstan reach an agreement to jointly build water-conservancy works over the Horgos River. |
| 1993/1/18 | CHN_KAZ | 4 | Water quantity | China and Kazakhstan sign an agreement to jointly construct a hydroelectric project on the Horgos River. The two sides decide to divide the construction costs. |
| 1999/5/5 | CHN_KAZ | 1 | Water quantity | Talks take place between China and Kazakhstan regarding problems of water intake from border rivers. |
| 1999/11/23 | CHN_KAZ | 2 | Water quantity | China and Kazakhstan sign the "Joint Communique of the People's Republic of China and the Republic of Kazakhstan on a Complete Resolution of All Border Issues". |
| 2001/3/24 | CHN_KAZ | 3 | Water quantity | Consultations between Kazakhstan and Chinese experts on the rational use of water resources of the transboundary rivers are conducted. |
| 2006/2/16 | CHN_KAZ | -1 | Water quantity | The Prime Minister of Kazakhstan acknowledges issues about the transboundary problem of the Irtysh and Ili rivers, and is unable to reach an agreement with China on the issues of environmental security. |

667