# Peer review of "Water resources management and dynamic changes in water politics"

_Hydrology and Earth System Sciences, 2020_

## Referee Comment (RC1) · Anonymous Referee #1 · 11 Feb 2021

General comments: This paper focuses on the development of water policies in the Central Asian (CA) transboundary rivers. Using the Gini Coefficient, the matching coefficient, the water conflict events, and the structure of water management institutions as indicators, this study reveals the complex management dynamics among the transboundary river basins in CA. The paper is generally well-written and structured, covering a broad range of data sources from both qualitative and quantitative perspectives. However, there are some issues that need to be addressed before acceptance.

Specific comments: Firstly, what are the major implications this article can deliver in reporting different perspectives of water policies development in the CA? The connections between the Gini coefficient, the matching coefficient, the number of water political events, and conflict/cooperative networks among the CA countries are not clear to me. One potential implication I can think of is that as the Gini coefficients and the matching coefficients indicate mismatches between water resources and socio-economic development, there is need to establish more cooperative network (rather than conflictive ones) among countries. A more elaborative discussion on how findings from the current situations in this paper should contribute to future management of transboundary rivers in the CA is needed.

Secondly, there is need for more justification about why these indicators are chosen in the method section. Why is the Gini Coefficient, combined with the matching coefficient good indicators for mismatches between water resources and socio-economic development? And how are changes of these coefficients impact on the water events? Are different countries showing different levels of impacts?

Thirdly, the flow among the three result sections should be strengthen. For example, what is the purpose of Section 3.1.1? I understand the authors want to provide a broad picture for the amount of water resources available in the CA river basins, but how this is connected to the remaining Sections 3.2 – 3.3 is not clear.

Technical corrections:

Line 110 onwards: There are brief introductions about the TFDD database but what about the World Water Conflict Chronology and the Interstate Commission for Water Coordination of Central Asia? Any issue when merging of data of different temporal periods?

Line 135 onwards: Clarifications about "what network" is needed: is the network only limited to among the five CA countries or other countries (as mentioned in Line 280) also included?

Line 347: Please clarify this sentence. Is water resources distribution unified in the

CA?

Line 626 (Figure 6): A timescale indicating which years these institutional changes occurred would be better.

Line 630 (Figure 7): It is clear that a single linear function is not suitable to represent the trend of the water events ($R^2$ only 0.02). I would recommend using step-wise regression function.

Line 641 onwards (Figure 9): it would be clearer for the readers if the same map scale is used across all four figures.

The whole paper needs to be grammatically checked again.

---

## Referee Comment (RC2) · Anonymous Referee #2 · 20 Feb 2021

General comments: This paper adopts three metrics to quantify the essential factors to drive water politics in the transboundary river basins of Centra Asia. The manuscript is organized logically and well written. The topic is relevant with the HESS audience and fall well within the scope of this special issue on transboundary river and socio-hydrology. However, the following comments should be addressed before its potential publication: 1) The Gini coefficient is traditionally used in economical discipline, which is calculated based on a large population (e.g., tens of millions). In this study, the coefficient is calculated based on 5 countries. Does that make sense to indicate the inequality issue? Actually, we can just compare water resource amount per land area / capita / etc among 5 CA countries to indicate their difference (or the inequality as

said by the authors). So what is the advantage of using Gini coefficient? Also, does that make sense to adopt the threshold value in Table 2 to evaluate inequality level of water issue? Similar concern is also applied to matching degree. As we have very limited country numbers in CA (i.e., 5), it is difficult to obtain a statistically meaningful coefficient. The authors need to demonstrate the rationality of adopted metrics and the threshold values. 2) For the water political event dataset, the authors combine different sources for different periods. The authors need to explain the consistency between TFDD, WWCC, and ICWCCA. 3) The authors are suggested to be careful with some conclusions, which should be drawn logically based on the supporting evidence. For example, in Sect 3.1.2, the authors conclude that "the quantity of water resources was not the causation of water contradictions in CA. Rather, the issues stemmed from the uneven allocation and utilization of water resources among these five countries". In the previous paragraphs, they discussed the mismatch between water and socio-economic elements including population, GDP, and cropland, but they did not discuss why water quantity is not an issue. Besides, at the end of discussion section, the authors discuss the approaches to eliminate conflicts and strength cooperation, which are useful but not logical in the context of research results. In discussion part, the readers may expect some logical deductions from the results, not just slogan.

Minor comments: 1) Ln32, use the latest number for transboundary rivers and other facts. The authors can refer to the papers in the same special issue. 2) Ln61, cite the original literature for the TFDD dataset. 3) Ln94, no rainfall feeds the river? 4) Ln122, what's n? 5) Ln169, release of water exceeds inflow, this confuses me. Especially when the authors say "since the Fergana Valley is an important agricultural region". Should not the agriculture consume a lot of water and cause release much lower than inflow? 6) Ln275, why include Tarim? Traditionally we do not consider Tarim as transboundary rivers. Maybe more specific to discuss Aksu? 7) Ln640, figure 9. The size of line is hard to differentiate as the number of water conflictive events. 8) Ln647, figure 10(b), the title of y-axis should be Number of water conflictive events? Check it.

---

## Author Comment (AC1) · 16 Mar 2021

Response to Reviewer #1

General comments: This paper focuses on the development of water policies in the Central Asian (CA) transboundary rivers. Using the Gini Coefficient, the matching coefficient, the water conflict events, and the structure of water management institutions as indicators, this study reveals the complex management dynamics among the transboundary river basins in CA. The paper is generally well-written and structured, covering a broad range of data sources from both qualitative and quantitative perspectives. However, there are some issues that need to be addressed before acceptance.

**-Author response:** We would like to thank reviewer #1 for the insightful comments and suggestions. These are very valuable to improve our manuscript, and the revised manuscript will follow the reviewer's recommendations. The followings are our point-by-point response to these comments.

Specific comments: Firstly, what are the major implications this article can deliver in reporting different perspectives of water policies development in the CA? The connections between the Gini coefficient, the matching coefficient, the number of water political events, and conflict/cooperative networks among the CA countries are not clear to me. One potential implication I can think of is that as the Gini coefficients and the matching coefficients indicate mismatches between water resources and socioeconomic development, there is need to establish more cooperative network (rather than conflictive ones) among countries. A more elaborative discussion on how findings from the current situations in this paper should contribute to future management of transboundary rivers in the CA is needed.

**-Author response:** Thanks for the instructive comment. We will add the implications of our findings to future management of transboundary rivers in CA in the revised manuscript. Firstly, just as you raised, based on the Gini coefficient and the matching coefficient of water and land resources, we have found that the matching degree of water and socio-economic elements (especially water and land resources) in CA is pretty poor. This is an important factor that increases the potential for water conflicts, and the main concern of the water conflictive events in CA is also the competitive utilization of water resources. Therefore, improving the water and land allocation systems and strengthening the water cooperative networks among countries will help reduce water conflicts and promote transboundary river management. Secondly, although there are more water cooperative events than conflictive events in CA, the cooperation mainly belong to weak levels based on our findings, and verbal supports (less effective) accounted for a large proportion (Level 1-2) in the current situation. There should be more high-level cooperation between the five countries, such as the military, economic or strategic supports, and freshwater treaties. The successful management of transboundary rivers in CA depends on deepening the countries' cooperation and trust. In addition, the CA should strengthen cooperation with its neighboring countries (such as Russia and China) in the water cooperative network, and make full use of the assistance of

international and regional organizations, because neighboring countries are the key trading partner and play an important role in the water policy reform of CA. We will add specific implications in the discussion part.

Secondly, there is need for more justification about why these indicators are chosen in the method section. Why is the Gini Coefficient, combined with the matching coefficient good indicators for mismatches between water resources and socio-economic development? And how are changes of these coefficients impact on the water events? Are different countries showing different levels of impacts?

**-Author response:** Thank you for the kind comment. We will explain the choosing criterion with more details in the method section. Gini coefficient is an objective indicator usually used to describe the degree of income distribution inequality. The distribution of water resources is not balanced in the region, which directly affects the agricultural production and economic development, and it is similar to the income distribution inequality. Therefore, the Gini coefficient has been effectively used as an indicator in measuring the degree of imbalance of water resources (Dai et al., 2018; Yan et al., 2016; Liu et al., 2018; Qin et al., 2020), and we used it to quantify the overall matching situation of water and socio-economic elements in CA. But the Gini coefficient cannot reflect the spatial differences among the five countries, so we combined the Gini coefficient with the matching coefficient of water and land resources, to represent the overall and individual matching degree of the five countries.

The matching situation of water resources and socio-economic elements (especially the land resources) in CA has an important impact on water politics. The higher the value of the Gini coefficient (or the smaller the matching coefficient of water and land resources), the worse the matching situation is, and the more likely the country will compete for water resources, so the greater the possibility of water conflictive events occur in the country. Conversely, the better the matching situation is, more water cooperative events may occur in the country. These coefficients are applicable to all five Central Asian countries and levels of impact are assumed to be the same. For example, we have found that Uzbekistan's water and land resources were poorly matched, and it is verified that the Uzbekistan was also at the core of the water conflictive network in our analysis in Section 3.3.3. Therefore, these coefficients effectively reflected the matching situation between water resources and socio-economic development, and were the prerequisite for analyzing the dynamics of water political events in CA. We will add more detailed analysis in the method section.

**References**
1.  Dai, C., Qin, X. S., Chen, Y., and Guo, H. C.: Dealing with equality and benefit for water allocation in a lake watershed: A Gini-coefficient based stochastic optimization approach, J. Hydrol., 561, 322-334, 2018.
2.  Liu, D., Liu, C. L., Fu, Q., Li, M., Faiz, M. A., Khan, M. I., Li, T. X., and Cui, S.: Construction and application of a refined index for measuring the regional matching

characteristics between water and land resources, Ecol. Indic., 91, 203-211, 2018.

3. Qin, J. N., Fu, X., and Peng, S. M.: Asymmetric benefit compensation model for resolving transboundary water management conflicts, Water Resour Manag., 34, 3625-3647, 2020.

4. Yan, F. Q., Zhang, S. W., Liu, X. T., Chen, D., Chen, J., Bu, K., Yang, J. C., and Chang, L. P.: The effects of spatiotemporal changes in land degradation on ecosystem services values in Sanjiang Plain, China, Remote Sens., 8(11), 917, 2016.

Thirdly, the flow among the three result sections should be strengthen. For example, what is the purpose of Section 3.1.1? I understand the authors want to provide a broad picture for the amount of water resources available in the CA river basins, but how this is connected to the remaining Sections 3.2 – 3.3 is not clear.

-**Author response:** Thank you for the instructive comment. We will strengthen the description of the connections among the three result sections in the revised manuscript. We realize that the purpose of Section 3.1.1 is not discussed in depth in the original version of the manuscript, and we will adjust the expression of this section. In fact, large reservoirs and dams occupy a key position in the water management infrastructure of CA and are vital to the economies of all five countries. The water stored in reservoirs is the main available water resources for agricultural irrigation and power generation in the river basins of CA. Therefore, analyzing the changing trends of inflow and outflow of large reservoirs can reflect the dynamics of available water resources and their development and utilization status, which is the basis for calculating the matching degree of water resources and socio-economic development in Section 3.1.2. On the other hand, humans play a leading role in the regulation and control of reservoirs in CA, and there is a competitive use between power generation in upstream countries and irrigation in downstream countries, so how will the allocation of water resources in reservoirs affect water conflicts and cooperation in the transboundary river basins of CA? Meanwhile, most of the reservoirs are aging and lack adequate maintenance, and the upgrading of water and energy facilities is one of the thorniest issues for the five Central Asian states, thereby what are challenges does it pose for water management in CA? Therefore, the analysis of large reservoirs on different sections of transboundary rivers reflected the ability of human to control water resources, and provides a precondition for the discussion of water management in Section 3.2 and water politics in Section 3.3. We will add more details about the connection in the result section.

Technical corrections:

Line 110 onwards: There are brief introductions about the TFDD database but what about the Water Conflict Chronology and the Interstate Commission for Water Coordination of Central Asia? Any issue when merging of data of different temporal periods?

-**Author response:** Thank you for the insightful comment. We realize that the descriptions of

the Water Conflict Chronology (WCC) and the Interstate Commission for Water Coordination of Central Asia (ICWCCA) databases are not detailed enough in the original manuscript. We will clarify these datasets and their data consistency in the revised manuscript.

Since both the water conflictive and cooperative events of CA recorded in the TFDD database occurred between 1951-2008, we selected the WCC database as the complementary data for water conflictive events and the ICWCCA database for water cooperative events during 2009-2018, respectively. The WCC is a detailed interactive online database that contains global conflicts over freshwater resources, where readers can retrieve and filter water conflicts by time, location, and subject (Gleick and Heberger, 2014). The data on water conflicts in CA cover the period during 1990-2018. To verify the consistency of conflictive events between TFDD and WCC, we compared the conflictive events from these two databases in their common timespan (1990-2008), and found that the conflictive events registered in the two datasets matches well with each other (Fig. s1a). The results indicated that the conflictive events by combining the TFDD and WCC database is reliable.

The ICWCCA is a joint committee established and authorized by the heads of the five Central Asian countries, which is responsible for making binding decisions on issues related to water distribution and utilization in the transboundary river basins of CA (Rahaman, 2012). It contains comprehensive water cooperative events such as conferences and agreements on transboundary rivers in CA since 2000. We also show that the two different datasets (TFDD and ICWCCA) produce similar trends of water cooperative events during 2000-2008 (Fig. s1b). This showed that the cooperative data obtained by merging the ICWCCA database and the TFDD is also reliable. Finally, we classified the levels of the complementary conflictive/cooperative events according to the classification criteria of water political events in TFDD.

[Figure]

**Figure 1:** Comparison of the number of water conflictive events in the TFDD and WCC datasets (a) and the number of water cooperative events in the TFDD and ICWCCA datasets (b)

**References**

1.   Gleick, P. H. and Heberger, M.: Water and conflict, in: The world's water, 159-171

January 2014, Washington, DC, Island Press, 2014.

2. Rahaman, M. M.: Principles of transboundary water resources management and water-related agreements in Central Asia: An analysis, Int. J. Water Resour. Dev., 28(3), 475-491, 2012.

Line 135 onwards: Clarifications about "what network" is needed: is the network only limited to among the five CA countries or other countries (as mentioned in Line 280) also included?

-**Author response:** Thank you for the helpful comment. We will explain it in the revised manuscript. The network covers all the countries that are involved in the water political events on the transboundary rivers of CA, and this network not only exists among the five Central Asian countries but also with other countries throughout the world that have cooperation or conflicts with CA.

Line 347: Please clarify this sentence. Is water resources distribution unified in the CA?

-**Author response:** Thank you for the insightful comment. We will explain it with more details in the revised manuscript. "water resources distribution unified" means that, CA's water resources were unified distribution by the Moscow government in the former Soviet Union. We realize that the original expression could not be clear, so the sentence will be changed to "water resources of transboundary rivers in CA have undergone the unified distribution during the former Soviet Union, and the separate management by the five Central Asian countries after the collapse".

Line 626 (Figure 6): A timescale indicating which years these institutional changes occurred would be better.

-**Author response:** Thank you for the kind suggestion. We will add the timescale in Figure 6 and improve the figure for a better presentation.

Line 630 (Figure 7): It is clear that a single linear function is not suitable to represent the trend of the water events (R^2 only 0.02). I would recommend using step-wise regression function.

-**Author response:** Thank you for the insightful comment. We will revise it in the revised manuscript. Yes, we realize that the single linear regression function has a low $R^2$ in fitting the trend of water events. The step-wise regression function can effectively retain the most significant independent variable through successive elimination, and needs multiple independent variables. In this study, the aim is to show the temporal trend of water events, so there is no need to apply the step-wise regression function. In fact, we have divided the study period into three stages to present the evolution of water events in the original manuscript: a stable period (1951-1991), rapid increase and decline (1991-2000), and a second stable period (2000-2018). Therefore, we consider it better to delete the single linear function and improve

the figure in the revised manuscript.

Line 641 onwards (Figure 9): It would be clearer for the readers if the same map scale is used across all four figures.

**-Author response:** Thank you the instructive comment. We will update and improve the figure to make it clear.

The whole paper needs to be grammatically checked again.

**-Author response:** Thank you for the insightful comment. We will check the grammar carefully and modify the errors. We will also improve the language by Halifax Proofreading and editing service (Canada).

---

## Author Comment (AC2) · 16 Mar 2021

Response to Reviewer #2

General comments: This paper adopts three metrics to quantify the essential factors to drive water politics in the transboundary river basins of Centra Asia. The manuscript is organized logically and well written. The topic is relevant with the HESS audience and fall well within the scope of this special issue on transboundary river and socio-hydrology. However, the following comments should be addressed before its potential publication:

**-Author response:** We would like to thank reviewer #2 for the instructive suggestions and comments. These are very valuable to improve our manuscript, and the revised manuscript will follow the reviewer's recommendations. Our explanations and responses to all the comments are listed below.

1) The Gini coefficient is traditionally used in economical discipline, which is calculated based on a large population (e.g., tens of millions). In this study, the coefficient is calculated based on 5 countries. Does that make sense to indicate the inequality issue? Actually, we can just compare water resource amount per land area / capita / etc among 5 CA countries to indicate their difference (or the inequality as said by the authors). So what is the advantage of using Gini coefficient? Also, does that make sense to adopt the threshold value in Table 2 to evaluate inequality level of water issue? Similar concern is also applied to matching degree. As we have very limited country numbers in CA (i.e., 5), it is difficult to obtain a statistically meaningful coefficient. The authors need to demonstrate the rationality of adopted metrics and the threshold values.

**-Author response:** Thanks for the instructive comment. We will clarify the rationality of the adopted metrics and the threshold values with more details in the revised manuscript. Gini coefficient is an objective indicator usually used to describe the degree of income distribution inequality. The distribution of water resources is not balanced in the region, which directly affects the agricultural production and economic development, and it is similar to the income distribution inequality. Therefore, the Gini coefficient has been effectively used as an indicator in measuring the degree of imbalance of water resources in countries or regions (e.g., in South Africa, Cole et al., 2018; in India, Malakar et al., 2018; the Sanjiang Plain in China, Yan et al., 2016; the Lake Dianchi Basin in China, Dai et al., 2018).

The higher the value of the Gini coefficient, the worse the matching situation is, and the more likely the country will compete for water resources. The amount of water resources per land area can show the relative spatiotemporal ratio between water resources and land resources, but it does not take into account the different types of water utilization. Thus, based on previous studies (Gunasekara et al., 2014; Yan et al., 2016; Dai et al., 2018; Qin et al., 2020), we used the Gini coefficient to comprehensively reflect the overall matching situation of water and socio-economic elements in CA, and analyzed the spatial matching differences among countries combined with the matching coefficient of water and land resources.

For the selection of the threshold values, the thresholds were widely recognized as effective in classification of matching degree between water resources and socio-economic development in many regions, including the CA (Yan et al., 2016; Liu et al., 2018; Zhang et al., 2020). We will clarify this issue in the revised method section.

**References**

1. Cole, M. J., Bailey, R. M., Cullis, J. D. S., and New, M. G.: Spatial inequality in water access and water use in South Africa, Water Policy, 20 (1), 37-52, 2018.
2. Dai, C., Qin, X. S., Chen, Y., and Guo, H. C.: Dealing with equality and benefit for water allocation in a lake watershed: A Gini-coefficient based stochastic optimization approach, J. Hydrol., 561, 322-334, 2018.
3. Gunasekara, N. K., Kazama, S., Yamazaki, D., and Oki, T.: Water conflict risk due to water resource availability and unequal distribution, Water Resour Manag., 28, 169-184, 2014.
4. Liu, D., Liu, C. L., Fu, Q., Li, M., Faiz, M. A., Khan, M. I., Li, T. X., and Cui, S.: Construction and application of a refined index for measuring the regional matching characteristics between water and land resources, Ecol. Indic., 91, 203-211, 2018.
5. Malakar, K., Mishra, T., and Patwardhan, A.: Inequality in water supply in India: an assessment using the Gini and Theil indices, Environ. Dev. Sustain., 20 (2), 841-864, 2018.
6. Qin, J. N., Fu, X., and Peng, S. M.: Asymmetric benefit compensation model for resolving transboundary water management conflicts, Water Resour Manag., 34, 3625-3647, 2020.
7. Yan, F. Q., Zhang, S. W., Liu, X. T., Chen, D., Chen, J., Bu, K., Yang, J. C., and Chang, L. P.: The effects of spatiotemporal changes in land degradation on ecosystem services values in Sanjiang Plain, China, Remote Sens., 8(11), 917, 2016.
8. Zhang, Y., Yan, Z. X., Song, J. X., and Wei, A. L.: Analysis for spatial-temporal matching pattern between water and land resources in Central Asia, Hydrol. Res., 51(5), 2020.

2) For the water political event dataset, the authors combine different sources for different periods. The authors need to explain the consistency between TFDD, WCC, and ICWCCA.

**-Author response:** Thank you for the helpful suggestion. We will explain about the consistency between TFDD, WCC, and ICWCCA in the revised manuscript. Since both the water conflictive and cooperative events of CA recorded in the TFDD database occurred between 1951-2008, we selected the WCC database as the complementary data for water conflictive events and the ICWCCA database for water cooperative events during 2009-2018, respectively. The WCC is a detailed interactive online database that contains global conflicts over freshwater resources, where readers can retrieve and filter water conflicts by time, location, and subject (Gleick and Heberger, 2014). The data on water conflicts in CA cover

the period during 1990-2018. To verify the consistency of conflictive events between TFDD and WCC, we compared the conflictive events from these two databases in their common timespan (1990-2008), and found that the conflictive events registered in the two datasets matches well with each other (Fig. s1a). The results indicated that the conflictive events by combining the TFDD and WCC database is reliable.

The ICWCCA is a joint committee established and authorized by the heads of the five Central Asian countries, which is responsible for making binding decisions on issues related to water distribution and utilization in the transboundary river basins of CA (Rahaman, 2012). It contains comprehensive water cooperative events such as conferences and agreements on transboundary rivers in CA since 2000. We also show that the two different datasets (TFDD and ICWCCA) produce similar trends of water cooperative events during 2000-2008 (Fig. s1b). This showed that the cooperative data obtained by merging the ICWCCA database and the TFDD is also reliable. Finally, we classified the levels of the complementary conflictive/cooperative events according to the classification criteria of water political events in TFDD.

[Figure]

**Figure 1:** Comparison of the number of water conflictive events in the TFDD and WCC datasets (a) and the number of water cooperative events in the TFDD and ICWCCA datasets (b)

**References**
1. Gleick, P. H. and Heberger, M.: Water and conflict, in: The world's water, 159-171 January 2014, Washington, DC, Island Press, 2014.
2. Rahaman, M. M.: Principles of transboundary water resources management and water-related agreements in Central Asia: An analysis, Int. J. Water Resour. Dev., 28(3), 475-491, 2012.

3) The authors are suggested to be careful with some conclusions, which should be drawn logically based on the supporting evidence. For example, in Sect 3.1.2, the authors conclude that "the quantity of water resources was not the causation of water contradictions in CA. Rather, the issues stemmed from the uneven allocation and utilization of water resources

among these five countries". In the previous paragraphs, they discussed the mismatch between water and socio-economic elements including population, GDP, and cropland, but they did not discuss why water quantity is not an issue. Besides, at the end of discussion section, the authors discuss the approaches to eliminate conflicts and strength cooperation, which are useful but not logical in the context of research results. In discussion part, the readers may expect some logical deductions from the results, not just slogan.

**-Author response:** Thank you for the insightful comment. We will strengthen the analysis to make all conclusions more clear and logically consistent. We will also explain why water quantity is not an issue in the revised manuscript. Water quantity refers to the total amount of water resources in CA, which equals to 3688.80 $m^3$ per capita and is in fact more than many regions of the world (e.g.,1148.00 $m^3$ per capita in India, 1989.33 $m^3$ per capita in China, and 3355.33 $m^3$ per capita in Japan). But the distribution of water resources among countries is extremely uneven. Kazakhstan has the largest amount of water resources ($643.50 \times 10^8$ $m^3$), followed by the upstream countries Tajikistan and Kyrgyzstan, which has $634.60 \times 10^8$ $m^3$ and $489.30 \times 10^8$ $m^3$, respectively. While the downstream countries Uzbekistan and Turkmenistan both have very few water resources, with $163.40 \times 10^8$ $m^3$ and $14.05 \times 10^8$ $m^3$, respectively (Wang et al., 2020). Therefore, the water contradictions in CA are not due to the shortage of total water quantity. Rather, the issues stemmed from the uneven allocation water resources and the mismatch between water and land resources among countries. We will add these analyses in the results section.

Meanwhile, we will revise the discussion section and propose the approaches to eliminate conflicts and strength cooperation based on our research results. The newly added discussion content may include the followings. Firstly, based on the Gini coefficient and the matching coefficient of water and land resources, we have found that the matching degree of water and socio-economic elements (especially water and land resources) in CA is pretty poor. This is an important factor that increases the potential for water conflicts, and the main concern of the water conflictive events in CA is also the competitive utilization of water resources. Therefore, improving the water and land allocation systems and strengthening the water cooperative networks among countries will help reduce water conflicts and promote transboundary river management. Secondly, although there are more water cooperative events than conflictive events in CA, the cooperation mainly belong to weak levels based on our findings, and verbal supports (less effective) accounted for a large proportion (Level 1-2) in the current situation. There should be more high-level cooperation between the five countries, such as the military, economic or strategic supports, and freshwater treaties. The successful management of transboundary rivers in CA depends on deepening the countries' cooperation and trust. In addition, the CA should strengthen cooperation with its neighboring countries (such as Russia and China) in the water cooperative network, and make full use of the assistance of international and regional organizations, because neighboring countries are the key trading partner and play an important role in the water policy reform of CA.

**Reference**

1. Wang, X. X., Chen, Y. N., Li, Z., Fang, G. H., and Wang, Y.: Development and utilization of water resources and assessment of water security in Central Asia, Agr. Water Manage., 240, 106297, 2020.

Minor comments:

Ln32, use the latest number for transboundary rivers and other facts. The authors can refer to the papers in the same special issue.

**-Author response:** Thank you for the instructive comment. We will revise and update it by referring to the papers in the same special issue.

Ln61, cite the original literature for the TFDD dataset.

**-Author response:** Thank you for the instructive comment. We will cite the original literature in the revised manuscript.

Ln94, no rainfall feeds the river?

**-Author response:** Thank you for the instructive comment. We will revise this sentence. There is little rainfall in CA, and the glaciers and snowmelt in high mountains account for a large share of the river's replenishment.

Ln122, what's $n$?

**-Author response:** Thank you for the insightful comment. The "$n$" represents the number of countries, and the value of "$n$" in this study is 5. We will explain it with more details in the revised manuscript.

Ln169, release of water exceeds inflow, this confuses me. Especially when the authors say "since the Fergana Valley is an important agricultural region". Should not the agriculture consume a lot of water and cause release much lower than inflow?

**-Author response:** Thank you for the insightful comment. To avoid any confusion, the sentence will be adjusted in the revised manuscript. In fact, the Andijan reservoir is located in mountainous areas and has no irrigation task of its own. Water entering the Andijan reservoir is mainly from the mountain rivers, and water released from the reservoir is most used for irrigation of agricultural areas in the Fergana Valley, downstream of the reservoir. Therefore, this may cause the release of water exceeds inflow in the Andijan reservoir.

Ln275, why include Tarim? Traditionally we do not consider Tarim as transboundary rivers. Maybe more specific to discuss Aksu?

**-Author response:** Thank you the instructive comment. We will add the explanation in the

revised manuscript. Traditionally we regard the Tarim River as an inland river in China, and the Aksu River, one of its sources, is a transboundary river. According to the latest version of TFDD in 2018 (McCracken and Wolf, 2019), Tarim as a transboundary river flows in China (area: 1048700 km$^2$, accounting for 95.5%), Kyrgyzstan (23900 km$^2$, 2.2%), Tajikistan (920 km$^2$, 0.1%), disputed area between India and China and administered by China (22200 km$^2$, 2.0%), disputed area between India and China and administered by India (2000 km$^2$, 0.2%), Kazakhstan (110 km$^2$, <0.1%). In addition, some scholars also regarded the Tarim as transboundary river in their studies (De Stefano et al., 2017; Yan et al., 2019). Therefore, we think it is appropriate to discuss the Tarim River.

**References**

1.  De Stefano, L., Petersen-Perlman, J. D., Sproles, E. A., Eynard, J., and Wolf, A. T.: Assessment of transboundary river basins for potential hydro-political tensions, Glob. Environ. Change-Human Policy, 45, 35-46, 2017.

2.  McCracken, M. and Wolf, A. T.: Updating the register of international river basins of the world, Int. J. Water Resour. Dev., 35(5), 732-777, 2019.

3.  Yan, J. B., Jia, S. F., Lv, A. F., and Zhu, W. B.: Water resources assessment of China's transboundary river basins using a machine learning approach, Water Resour. Res., 55(1), 632-655, 2019.

Ln640, figure 9. The size of line is hard to differentiate as the number of water conflictive events.

**-Author response:** Thank you for the kind comment. We will revise and update the figure to make it clear.

Ln647, figure 10(b), the title of y-axis should be Number of water conflictive events? Check it.

**-Author response:** Thank you the instructive comment. We will check it. A total of 53 water conflictive events were recorded in CA, of which the most occurred in January (9 conflictive events). We want to show the monthly distribution of water conflictive events, so the title of y-axis is the number of water conflictive events, and the title of x-axis is month.

---

## Author Response (AR1)

Dear Dr. Jing Wei,

On behalf of my co-authors, we thank you very much for giving us an opportunity to revise our manuscript. We appreciate you and the two anonymous reviewers for the instructive and helpful comments. These comments are very substantial to improve the quality of our manuscript.

We have revised the manuscript based on the reviewer's recommendations. Following this letter are the reviewer's comments and our point-by-point response, including all relevant changes made in the manuscript.

Thank you and best regards.

Yours sincerely,

Zhi Li

Response to Reviewer #1

General comments: This paper focuses on the development of water policies in the Central Asian (CA) transboundary rivers. Using the Gini Coefficient, the matching coefficient, the water conflict events, and the structure of water management institutions as indicators, this study reveals the complex management dynamics among the transboundary river basins in CA. The paper is generally well-written and structured, covering a broad range of data sources from both qualitative and quantitative perspectives. However, there are some issues that need to be addressed before acceptance.

**-Author response:** We would like to thank reviewer #1 for the insightful comments and suggestions. These are very valuable to improve the quality of our manuscript. All the comments have been taken into account carefully and we have revised the manuscript accordingly. The followings are our point-by-point response and revisions to these comments.

Specific comments: Firstly, what are the major implications this article can deliver in reporting different perspectives of water policies development in the CA? The connections between the Gini coefficient, the matching coefficient, the number of water political events, and conflict/cooperative networks among the CA countries are not clear to me. One potential implication I can think of is that as the Gini coefficients and the matching coefficients indicate mismatches between water resources and socioeconomic development, there is need to establish more cooperative network (rather than conflictive ones) among countries. A more elaborative discussion on how findings from the current situations in this paper should contribute to future management of transboundary rivers in the CA is needed.

**-Author response:** Thank you for the instructive suggestion. We have added the implications of our findings for future management of transboundary rivers in CA in the revised manuscript (Lines 425-437).

*Lines 425-437:*

*From our findings, we draw the following implications for eliminating conflicts and strengthening future cooperation in the transboundary rivers of CA. Firstly, as both the Gini coefficient and the matching coefficient of water and land resources indicate, the matching between water and socio-economic elements (especially land resources) in CA is pretty poor. This mismatch increases the potential for water conflicts, and the primary concern of water conflictive events in CA is also the competitive utilization of water resources. Therefore, improving the water and land allocation systems and strengthening the water cooperative networks between countries will help reduce water conflicts and promote transboundary river management in the region. Secondly, although there are more water cooperative events than conflictive events in CA, the cooperation is mainly low-level based on our findings, and verbal supports (less effective) account for a large proportion (level 1-2) in the current situation. There should be more high-level cooperation among the five countries, such as the*

*military, economic or strategic supports, and freshwater treaties. The successful management of transboundary rivers in CA depends on deepening the countries' cooperation and trust. In addition, CA should make utilize the assistance of international and regional organizations, and enhance cooperation with its neighboring countries (such as Russia and China), as these neighboring countries are CA's key trading partners and play an important role in water policy reform in the region.*

Secondly, there is need for more justification about why these indicators are chosen in the method section. Why is the Gini Coefficient, combined with the matching coefficient good indicators for mismatches between water resources and socio-economic development? And how are changes of these coefficients impact on the water events? Are different countries showing different levels of impacts?

**-Author response:** Thank you for the kind comment. We have explained the choosing criterion with more details in the revised manuscript. Gini coefficient is an objective indicator usually used to quantify the inequality of income distribution. The distribution of water resources is uneven in the region, which directly affects the agricultural production and economic development, and it is similar to the income distribution inequality. Therefore, the Gini coefficient has been used as an effective indicator of the degree of imbalance in water resources between countries or regions (Yan et al., 2016; Dai et al., 2018; Liu et al., 2018; Qin et al., 2020). Other indicators, such as the amount of water resources per land area, per capita, etc., can reveal spatiotemporal relationships between water resources and socioeconomic factors, but they do not take into account the different types of water utilization. Thus, in this study, we used the Gini coefficient to quantify the overall matching between water and socio-economic factors in CA. Due to the Gini coefficient alone cannot reflect spatial differences among the five countries, we combined the Gini coefficient with the matching coefficient of water and land resources, to represent the overall and individual matching degree of the five countries. We have added detailed description in the method section (Lines 131-137; Lines 151-152).

The matching situation between water resources and socio-economic elements (especially the land resources) in CA has an important impact on water politics. The higher the value of the Gini coefficient (or the smaller the matching coefficient of water and land resources), the lower the degree of matching, and the higher the likelihood of competition for water resources in the region, so the greater the possibility of water conflictive events. Conversely, the higher the degree of matching, the lower the possibility of water conflictive events in the region. These coefficients are applicable to all five Central Asian countries and levels of impact are assumed to be the consistent. For example, we have found that Uzbekistan's water and land resources were poorly matched, and it is verified that the Uzbekistan was also at the core of water conflictive network in our analysis in Section 3.3.3. Therefore, these coefficients effectively reflect the matching situation between water resources and socio-economic

development, and understanding the current status of the matching situation is the prerequisite for analyzing the dynamics of water political events in CA. We have clarified these coefficients in the method section (Lines 138-142).

*Lines 131-137:*

*The Gini coefficient is an economic index proposed by the Italian economist Corrado Gini to quantify the inequality of income distribution (Shlomo, 1979). The distribution of water resources is uneven in the region, which directly affects the agricultural production and economic development, and it is similar to the income distribution inequality. For this reason, the Gini coefficient has been used as an effective indicator of the degree of imbalance in water resources between countries or regions (e.g., South Africa, Cole et al., 2018; India, Malakar et al., 2018; the Sanjiang Plain in China, Yan et al., 2016; the Lake Dianchi Basin in China, Dai et al., 2018), and we use the Gini coefficient in this study to quantify the overall matching between water and socio-economic factors in CA.*

*Lines 151-152:*

*As the Gini coefficient cannot reflect spatial variations between countries, we use the matching coefficient of water and land resources to represent the individual matching degree of the five countries.*

*Lines 138-142:*

*The value of the Gini coefficient ranges between 0 and 1. The closer it is to 1, the lower the degree of matching, and the higher the likelihood of competition for water resources in the region, so the greater the possibility of water conflictive events; conversely, the closer it is to 0, the higher the degree of matching, and the lower the possibility of water conflictive events in the region. The Gini coefficient is applicable to all five Central Asian countries, and the level of impact is assumed to be the consistent.*

**References**

1. Cole, M. J., Bailey, R. M., Cullis, J. D. S., and New, M. G.: Spatial inequality in water access and water use in South Africa, Water Policy, 20 (1), 37-52, 2018.

2. Dai, C., Qin, X. S., Chen, Y., and Guo, H. C.: Dealing with equality and benefit for water allocation in a lake watershed: A Gini-coefficient based stochastic optimization approach, J. Hydrol., 561, 322-334, 2018.

3. Liu, D., Liu, C. L., Fu, Q., Li, M., Faiz, M. A., Khan, M. I., Li, T. X., and Cui, S.: Construction and application of a refined index for measuring the regional matching characteristics between water and land resources, Ecol. Indic., 91, 203-211, 2018.

4. Malakar, K., Mishra, T., and Patwardhan, A.: Inequality in water supply in India: an assessment using the Gini and Theil indices, Environ. Dev. Sustain., 20 (2), 841-864, 2018.

5. Qin, J. N., Fu, X., and Peng, S. M.: Asymmetric benefit compensation model for resolving transboundary water management conflicts, Water Resour Manag., 34, 3625-3647, 2020.

6.  Shlomo, Y.: Relative deprivation and the Gini coefficient, Q. J. Econ., 93(2), 321-324, 1979.
7.  Yan, F. Q., Zhang, S. W., Liu, X. T., Chen, D., Chen, J., Bu, K., Yang, J. C., and Chang, L. P.: The effects of spatiotemporal changes in land degradation on ecosystem services values in Sanjiang Plain, China, Remote Sens., 8(11), 917, 2016.

Thirdly, the flow among the three result sections should be strengthen. For example, what is the purpose of Section 3.1.1? I understand the authors want to provide a broad picture for the amount of water resources available in the CA river basins, but how this is connected to the remaining Sections 3.2 – 3.3 is not clear.

**-Author response:** Thank you for the instructive comment. We have strengthened the description of the connections among three result sections in the revised manuscript. We realize that the purpose of Section 3.1.1 was not discussed in depth in the original version of the manuscript, and we have adjusted the expression of this section to address the power of water manipulation in CA. In fact, large reservoirs and dams occupy a key position in the water infrastructure management of CA and are vital to the economies of all five countries. The water contained in reservoirs is the primary freshwater resource in the region's transboundary river basins. Therefore, analyzing the changing trends in the inflow and outflow of large reservoirs can reflect the dynamics and utilization of available water resources in CA. Meanwhile, Humans play a leading role in the operational regulation and control of these reservoirs, and there is a competitive water use between power generation in upstream countries and agricultural irrigation in downstream countries. Thus, the allocation of the water resources in reservoirs is a key factor influencing water conflicts and cooperation in the transboundary river basins of CA, which is the basis for water political events analysis in Section 3.3. We have also found in Section 3.3 that the construction and development of reservoirs are the second major theme of water political events in CA.

Additionally, most dams and reservoirs in CA are aging and lack of adequate maintenance, and the upgrading of water and energy facilities is one of the most contentious issues for the five Central Asian states. This poses great challenges for water management in CA as we mentioned in Section 3.2. Therefore, the analysis of large reservoirs on different sections of transboundary rivers provides a precondition for the discussion of water management in Section 3.2 and water politics in Section 3.3. We have added more details about the connection in the result section (Lines 183-190; Lines 209-214).

*Lines 183-190:*
*Large reservoirs and dams occupy a key position in the water infrastructure management of CA and are vital to the economies of all five countries. More than 290 reservoirs with a total storage capacity of 163.19 $km^3$ exist in CA. The water contained in reservoirs is the primary freshwater resource in the region's transboundary river basins, and the changing trends in the inflow and outflow of large reservoirs reflect the dynamics and utilization of available*

*water resources in CA. Humans play a leading role in the operational regulation and control of these reservoirs, and there is a competitive water use between power generation in upstream countries and agricultural irrigation in downstream countries. Therefore, the allocation of the water resources in reservoirs is a key factor influencing water conflicts and cooperation in the transboundary river basins of CA.*

**Lines 209-214:**

*Additionally, most dams and reservoirs in CA are aging and lack of adequate maintenance, or even with insufficient funds to maintain normal operation. This situation, coupled with the increasing population in the floodplain downstream, significantly increases the water resource risk in the region. One outcome of this risk was the 2010 flooding in Kazakhstan, caused by the collapse of the Kyzyl-Agash Dam (Libert and Lipponen, 2012). In general, the upgrading of water and energy facilities is one of the most contentious issues for the five Central Asian states and poses significant challenges to water management in CA.*

**Reference**

1. Libert, B. O. and Lipponen, A.: Challenges and opportunities for transboundary water cooperation in Central Asia: findings from UNECE's Regional Assessment and Project Work, Int. J. Water Resour. Dev., 28(3), 565-576, 2012.

Technical corrections:

Line 110 onwards: There are brief introductions about the TFDD database but what about the Water Conflict Chronology and the Interstate Commission for Water Coordination of Central Asia? Any issue when merging of data of different temporal periods?

**-Author response:** Thank you for the insightful comment. We realize that the descriptions of the Water Conflict Chronology (WCC) and the Interstate Commission for Water Coordination of Central Asia (ICWCCA) databases are not detailed enough in the original manuscript. We have clarified these datasets and their data consistency in the revised manuscript (Lines 112-127; Lines 638-640).

**Lines 112-127:**

*Since the TFDD database only documents events of water conflict and cooperation during the 1951-2008 period, for the 2009-2018 period, we used water conflictive events from the Water Conflict Chronology (WCC) database and water cooperative events from the Interstate Commission for Water Coordination of Central Asia (ICWCCA) database. The WCC is a detailed interactive online database that contains global conflicts over freshwater resources (https://www.worldwater.org/water-conflict/) (Gleick and Heberger, 2014). The WCC data can be retrieved and filtered according to time, location and subject, and the data on water conflict in CA cover the period during 1990-2018. To verify the consistency of conflictive events between TFDD and WCC, we compared the conflictive events registered in the two databases for their common timespan (1990-2008). The events concurred with each other*

*(Fig. 3a), confirming that the conflictive events obtained by combining the TFDD and WCC databases were reliable.*

*The ICWCCA is a joint committee established and authorized by the heads of the five Central Asian countries (http://www.icwc-aral.uz/), which is responsible for making binding decisions on issues related to water distribution and utilization in the transboundary river basins of CA (Rahaman, 2012). It contains comprehensive records of water cooperative events, such as conferences and agreements on transboundary rivers in CA, from 2000 onwards. The TFDD and ICWCCA datasets indicated similar trends of water cooperative events during the 2000-2008 period, the common timespan of the two datasets (Fig. 3b), confirming that the cooperative events obtained by merging the TFDD and ICWCCA databases were also reliable.*

*Lines 638-640:*

[Figure]

*Figure 3: Comparison of the number of water conflictive events in the TFDD and WCC datasets (a) and the number of water cooperative events in the TFDD and ICWCCA datasets (b)*

**References**

1.  Gleick, P. H. and Heberger, M.: Water and conflict, in: The world's water, 159-171 January 2014, Washington, DC, Island Press, 2014.
2.  Rahaman, M. M.: Principles of transboundary water resources management and water-related agreements in Central Asia: An analysis, Int. J. Water Resour. Dev., 28(3), 475-491, 2012.

Line 135 onwards: Clarifications about "what network" is needed: is the network only limited to among the five CA countries or other countries (as mentioned in Line 280) also included?

**-Author response:** Thank you for the insightful comment. We have explained it in the revised manuscript (Lines 166-168).

*Lines 166-168:*
*The network comprises all the countries that are involved in water political events over CA's*

*transboundary rivers. In addition to the five Central Asian countries, the network includes any other country that cooperates or clashes with Central Asian countries over water resources.*

Line 347: Please clarify this sentence. Is water resources distribution unified in the CA?

**-Author response:** Thank you for the insightful comment. "Water resources distribution unified" means that, CA's water resources were unified distribution by the Moscow government in the former Soviet Union. We realize that the original expression could not be clear, and we have adjusted the sentence in the revised manuscript (Lines 382-383).

*Lines 382-383:*

*The water resources of CA's transboundary rivers underwent a unified distribution during the former Soviet Union, and separate management by the five Central Asian countries after its collapse.*

Line 626 (Figure 6): A timescale indicating which years these institutional changes occurred would be better.

**-Author response:** Thank you for the kind comment. For a better presentation, we have added the years in which major institutional changes occurred and improved the figure in the revised manuscript (Lines 653-655).

*Lines 653-655:*

[Figure]

*Figure 7: Evolution of water management policies and institutional framework in Central Asia.*

*Note: The numbers in red are the years in which major institutional changes occurred.*

Line 630 (Figure 7): It is clear that a single linear function is not suitable to represent the trend of the water events (R^2 only 0.02). I would recommend using step-wise regression function.

**-Author response:** Thank you for the insightful comment. We have revised it. Yes, we realize that the single linear regression function has a low $R^2$ in fitting the trend of water events. The step-wise regression function can effectively retain the most significant independent variable through successive elimination, and needs multiple independent variables. In this study, the aim is to show the temporal trend of water events, so there is no need to apply the step-wise regression function. In fact, we have divided the study period into three stages to present the evolution of water events in the original manuscript: a stable period (1951-1991), a rapid increase and decline period (1991-2001), and a second stable period (2001-2018). Therefore, we have deleted the single linear function and improved the figure in the revised manuscript (Lines 657-659).

*Lines 657-659:*

[Figure]

*Figure 8: Changing trends in water conflictive, cooperative and total water political events in Central Asia from 1951 to 2018.*

*Note: P1- a stable period; P2- a rapid increase and decline period; P3- a second stable period.*

Line 641 onwards (Figure 9): It would be clearer for the readers if the same map scale is used across all four figures.

**-Author response:** Thank you for the instructive comment. We have adjusted the map scale and improved the figure in the revised manuscript (Lines 667-670).

*Lines 667-670:*

[Figure]

*Figure 10: Water conflictive and cooperative networks between Central Asian countries and other countries in the world: (a) Number of water conflictive events in 1951-1991 and (b) 1992-2018; (c) number of water cooperative events in 1951-1991 and (d) 1992-2018.*

The whole paper needs to be grammatically checked again.

**-Author response:** Thank you for the insightful comment. We have checked the grammar carefully and modified the errors. We have also improved the language by Halifax Proofreading and editing service (Canada).

Response to Reviewer #2

General comments: This paper adopts three metrics to quantify the essential factors to drive water politics in the transboundary river basins of Centra Asia. The manuscript is organized logically and well written. The topic is relevant with the HESS audience and fall well within the scope of this special issue on transboundary river and socio-hydrology. However, the following comments should be addressed before its potential publication:

-**Author response:** We would like to thank reviewer #2 for the instructive suggestions and comments. These are very substantial to improve our manuscript, and the revised manuscript is based on the reviewer's recommendations. Our response and revisions to all the comments are listed below.

1) The Gini coefficient is traditionally used in economical discipline, which is calculated based on a large population (e.g., tens of millions). In this study, the coefficient is calculated based on 5 countries. Does that make sense to indicate the inequality issue? Actually, we can just compare water resource amount per land area / capita / etc among 5 CA countries to indicate their difference (or the inequality as said by the authors). So what is the advantage of using Gini coefficient? Also, does that make sense to adopt the threshold value in Table 2 to evaluate inequality level of water issue? Similar concern is also applied to matching degree. As we have very limited country numbers in CA (i.e., 5), it is difficult to obtain a statistically meaningful coefficient. The authors need to demonstrate the rationality of adopted metrics and the threshold values.

-**Author response:** Thanks for the instructive comment. We have clarified the rationality of adopted metrics and the threshold values with more details in the revised manuscript. Gini coefficient is an objective indicator usually used to quantify the inequality of income distribution. The distribution of water resources is uneven in the region, which directly affects the agricultural production and economic development, and it is similar to the income distribution inequality. For this reason, the Gini coefficient has been used as an effective indicator of the degree of imbalance in water resources between countries or regions (e.g., South Africa, Cole et al., 2018; India, Malakar et al., 2018; the Sanjiang Plain in China, Yan et al., 2016; the Lake Dianchi Basin in China, Dai et al., 2018).

Other indicators, such as the amount of water resources per land area, per capita, etc., can reveal spatiotemporal relationships between water resources and socioeconomic factors, but they do not take into account the different types of water utilization. Thus, based on previous studies (Gunasekara et al., 2014; Yan et al., 2016; Dai et al., 2018; Qin et al., 2020), we used the Gini coefficient to quantify the overall matching between water and socio-economic factors in CA, and analyzed the spatial matching differences among countries combined with the matching coefficient of water and land resources. The higher the value of the Gini coefficient, the lower the degree of matching, and the higher the likelihood of competition for

water resources in the region, so the greater the possibility of water conflictive events. Conversely, the higher the degree of matching, the lower the possibility of water conflictive events in the region. We have added the analysis in the method section (Lines 131-141).

*Lines 131-141:*

*The Gini coefficient is an economic index proposed by the Italian economist Corrado Gini to quantify the inequality of income distribution (Shlomo, 1979). The distribution of water resources is uneven in the region, which directly affects the agricultural production and economic development, and it is similar to the income distribution inequality. For this reason, the Gini coefficient has been used as an effective indicator of the degree of imbalance in water resources between countries or regions (e.g., South Africa, Cole et al., 2018; India, Malakar et al., 2018; the Sanjiang Plain in China, Yan et al., 2016; the Lake Dianchi Basin in China, Dai et al., 2018), and we use the Gini coefficient in this study to quantify the overall matching between water and socio-economic factors in CA.*

*The value of the Gini coefficient ranges between 0 and 1. The closer it is to 1, the lower the degree of matching, and the higher the likelihood of competition for water resources in the region, so the greater the possibility of water conflictive events; conversely, the closer it is to 0, the higher the degree of matching, and the lower the possibility of water conflictive events in the region.*

For the rationality of the threshold values, we have clarified this issue in the revised method section (Lines 148-149).

*Lines 148-149:*

*These thresholds are widely acknowledged to be effective in classifying the matching degree between water resources and socio-economic development in many regions with small samples (Yan et al., 2016; Liu et al., 2018).*

**References**
1. Cole, M. J., Bailey, R. M., Cullis, J. D. S., and New, M. G.: Spatial inequality in water access and water use in South Africa, Water Policy, 20 (1), 37-52, 2018.
2. Dai, C., Qin, X. S., Chen, Y., and Guo, H. C.: Dealing with equality and benefit for water allocation in a lake watershed: A Gini-coefficient based stochastic optimization approach, J. Hydrol., 561, 322-334, 2018.
3. Gunasekara, N. K., Kazama, S., Yamazaki, D., and Oki, T.: Water conflict risk due to water resource availability and unequal distribution, Water Resour Manag., 28, 169-184, 2014.
4. Liu, D., Liu, C. L., Fu, Q., Li, M., Faiz, M. A., Khan, M. I., Li, T. X., and Cui, S.: Construction and application of a refined index for measuring the regional matching characteristics between water and land resources, Ecol. Indic., 91, 203-211, 2018.
5. Malakar, K., Mishra, T., and Patwardhan, A.: Inequality in water supply in India: an assessment using the Gini and Theil indices, Environ. Dev. Sustain., 20 (2), 841-864,

2018.

6. Qin, J. N., Fu, X., and Peng, S. M.: Asymmetric benefit compensation model for resolving transboundary water management conflicts, Water Resour Manag., 34, 3625-3647, 2020.

7. Shlomo, Y.: Relative deprivation and the Gini coefficient, Q. J. Econ., 93(2), 321-324, 1979.

8. Yan, F. Q., Zhang, S. W., Liu, X. T., Chen, D., Chen, J., Bu, K., Yang, J. C., and Chang, L. P.: The effects of spatiotemporal changes in land degradation on ecosystem services values in Sanjiang Plain, China, Remote Sens., 8(11), 917, 2016.

2) For the water political event dataset, the authors combine different sources for different periods. The authors need to explain the consistency between TFDD, WCC, and ICWCCA.

**-Author response:** Thank you for the helpful suggestion. We have explained about the consistency between TFDD, WCC, and ICWCCA in the revised manuscript (Lines 112-127; Lines 638-640).

*Lines 112-127:*

*Since the TFDD database only documents events of water conflict and cooperation during the 1951-2008 period, for the 2009-2018 period, we used water conflictive events from the Water Conflict Chronology (WCC) database and water cooperative events from the Interstate Commission for Water Coordination of Central Asia (ICWCCA) database. The WCC is a detailed interactive online database that contains global conflicts over freshwater resources (https://www.worldwater.org/water-conflict/) (Gleick and Heberger, 2014). The WCC data can be retrieved and filtered according to time, location and subject, and the data on water conflict in CA cover the period during 1990-2018. To verify the consistency of conflictive events between TFDD and WCC, we compared the conflictive events registered in the two databases for their common timespan (1990-2008). The events concurred with each other (Fig. 3a), confirming that the conflictive events obtained by combining the TFDD and WCC databases were reliable.*

*The ICWCCA is a joint committee established and authorized by the heads of the five Central Asian countries (http://www.icwc-aral.uz/), which is responsible for making binding decisions on issues related to water distribution and utilization in the transboundary river basins of CA (Rahaman, 2012). It contains comprehensive records of water cooperative events, such as conferences and agreements on transboundary rivers in CA, from 2000 onwards. The TFDD and ICWCCA datasets indicated similar trends of water cooperative events during the 2000-2008 period, the common timespan of the two datasets (Fig. 3b), confirming that the cooperative events obtained by merging the TFDD and ICWCCA databases were also reliable.*

*Lines 638-640:*

[Figure]

*Figure 3: Comparison of the number of water conflictive events in the TFDD and WCC datasets (a) and the number of water cooperative events in the TFDD and ICWCCA datasets (b)*

**References**

1. Gleick, P. H. and Heberger, M.: Water and conflict, in: The world's water, 159-171 January 2014, Washington, DC, Island Press, 2014.
2. Rahaman, M. M.: Principles of transboundary water resources management and water-related agreements in Central Asia: An analysis, Int. J. Water Resour. Dev., 28(3), 475-491, 2012.

3) The authors are suggested to be careful with some conclusions, which should be drawn logically based on the supporting evidence. For example, in Sect 3.1.2, the authors conclude that "the quantity of water resources was not the causation of water contradictions in CA. Rather, the issues stemmed from the uneven allocation and utilization of water resources among these five countries". In the previous paragraphs, they discussed the mismatch between water and socio-economic elements including population, GDP, and cropland, but they did not discuss why water quantity is not an issue. Besides, at the end of discussion section, the authors discuss the approaches to eliminate conflicts and strength cooperation, which are useful but not logical in the context of research results. In discussion part, the readers may expect some logical deductions from the results, not just slogan.

**-Author response:** Thank you for the insightful comment. We have strengthened the analysis to make all conclusions clear and logically consistent in the revised manuscript. "Water quantity" refers to the total amount of water resources in CA, and we have explained why water quantity is not an issue in the results section (Lines 239-247). Meanwhile, we have adjusted the discussion section and proposed corresponding approaches to eliminate conflicts and strength cooperation based on our research results (Lines 425-437).

*Lines 239-247:*

*In fact, the amount of water resources in CA is relatively abundant, which equals to 3688.80 $m^3$ per capita and is more than many regions of the world (e.g.,1148.00 $m^3$ per capita in India, 1989.33 $m^3$ per capita in China, and 3355.33 $m^3$ per capita in Japan). The distribution of*

*water resources among the Central Asian countries, however, is extremely uneven. Kazakhstan has the largest amount of water resources (643.50×108 m3), followed by the upstream countries of Tajikistan and Kyrgyzstan (634.60×108 m3 and 489.30×108 m3, respectively). While the downstream countries, Uzbekistan and Turkmenistan, have scarce water resource (163.40×108 m3 and 14.05×108 m3, respectively) (Wang et al., 2020a). Therefore, the water contradictions in CA are not straightly caused by the shortage of total water quantity. Rather, from the above analysis, the issues could be attributed to the uneven allocation water resources and the mismatch between water and land resources among the Central Asian countries (Chen et al., 2018).*

***Lines 425-437:***

*From our findings, we draw the following implications for eliminating conflicts and strengthening future cooperation in the transboundary rivers of CA. Firstly, as both the Gini coefficient and the matching coefficient of water and land resources indicate, the matching between water and socio-economic elements (especially land resources) in CA is pretty poor. This mismatch increases the potential for water conflicts, and the primary concern of water conflictive events in CA is also the competitive utilization of water resources. Therefore, improving the water and land allocation systems and strengthening the water cooperative networks between countries will help reduce water conflicts and promote transboundary river management in the region. Secondly, although there are more water cooperative events than conflictive events in CA, the cooperation is mainly low-level based on our findings, and verbal supports (less effective) account for a large proportion (level 1-2) in the current situation. There should be more high-level cooperation among the five countries, such as the military, economic or strategic supports, and freshwater treaties. The successful management of transboundary rivers in CA depends on deepening the countries' cooperation and trust. In addition, CA should make utilize the assistance of international and regional organizations, and enhance cooperation with its neighboring countries (such as Russia and China), as these neighboring countries are CA's key trading partners and play an important role in water policy reform in the region.*

**References**

1. Chen, Y. N., Li, Z., Fang, G. H., and Li, W. H.: Large hydrological processes changes in the transboundary rivers of Central Asia, J. Geophys. Res. Atmos., 123 (10), 5059-5069, 2018.
2. Wang, X. X., Chen, Y. N., Li, Z., Fang, G. H., and Wang, Y.: Development and utilization of water resources and assessment of water security in Central Asia, Agr. Water Manage., 240, 106297, 2020.

Minor comments:

Ln32, use the latest number for transboundary rivers and other facts. The authors can refer to the papers in the same special issue.

**-Author response:** Thank you for the instructive comment. We have revised and updated the numbers by referring to the papers in the same special issue (Lines 31-32).

*Lines 31-32:*
*There are 310 transboundary rivers worldwide involving 150 countries, ... (Di Baldassarre et al., 2013; McCracken and Wolf, 2019; Wei et al., 2021).*

**References**

1. Di Baldassarre, G., Viglione, A., Carr, G., Kuil, L., Salinas, J., and Blöschl, G.: Socio-hydrology: conceptualising human-flood interactions, Hydrol. Earth Syst. Sci., 17, 3295, 2013.
2. McCracken, M. and Wolf, A. T.: Updating the register of international river basins of the world, Int. J. Water Resour. Dev., 35(5), 732-777, 2019.
3. Wei, J., Wei, Y., Tian, F., Nott, N., de Witt, C., Guo, L., and Lu, Y.: News media coverage of conflict and cooperation dynamics of water events in the Lancang-Mekong River basin, Hydrol. Earth Syst. Sci., 25, 1603-1615, 2021.

Ln61, cite the original literature for the TFDD dataset.

**-Author response:** Thank you for the instructive comment. We have changed accordingly in the revised manuscript (Lines 59-60).

*Line 59-60:*
*The Transboundary Freshwater Dispute Database (TFDD), established by researchers at Oregon State University (Wolf, 1999).*

**Reference**

1. Wolf, A. T.: The Transboundary Freshwater Dispute Database project, Water Int., 24(2), 160-163, 1999.

Ln94, no rainfall feeds the river?

**-Author response:** Thank you for the instructive comment. There is little rainfall in CA, and the glaciers and snowmelt in high mountains account for a large share of the river's replenishment (Chen et al., 2018). We have revised this sentence (Lines 93-94).

*Lines 93-94:*
*..., and mainly supplied by snowmelt, glaciers and precipitation.*

**Reference**

1. Chen, Y. N., Li, Z., Fang, G. H., and Li, W. H.: Large hydrological processes changes in the transboundary rivers of Central Asia, J. Geophys. Res. Atmos., 123 (10), 5059-5069, 2018.

Ln122, what's *n*?

**-Author response:** Thank you for the insightful comment. The "*n*" represents the number of countries, and the value of "*n*" in this study is 5. We have explained it in the revised manuscript (Line 145).

*Line 145:*

*…, n represents the number of countries (in this study, n = 5).*

Ln169, release of water exceeds inflow, this confuses me. Especially when the authors say "since the Fergana Valley is an important agricultural region". Should not the agriculture consume a lot of water and cause release much lower than inflow?

**-Author response:** Thank you for the insightful comment. To avoid any confusion, the sentence has been adjusted in the revised manuscript (Lines 196-199). In fact, the Andijan reservoir is located in mountainous areas and has no irrigation task of its own. Water entering the Andijan reservoir is mainly from alpine rivers, and water released from the reservoir is most used for irrigation of agricultural areas in the Fergana Valley, downstream of the reservoir. Therefore, this may cause the release of the Andijan reservoir higher than inflow.

*Lines 196-199:*

*The Andijan reservoir is located on the Kara Darya River, in the upper reaches of the Fergana Valley (an agricultural area of regional importance). From 2010 to 2017, the Andijan reservoir received an average inflow of 4.82 $km^3$/a, primarily from alpine rivers. The average outflow recorded was 5.34 $km^3$/a, and most of the released water was used for crop irrigation in the Fergana Valley.*

Ln275, why include Tarim? Traditionally we do not consider Tarim as transboundary rivers. Maybe more specific to discuss Aksu?

**-Author response:** Thank you for the instructive comment. Traditionally we regard the Tarim River as an inland river in China, and the Aksu River, one of its main sources, is a transboundary river. According to the latest version of TFDD in 2018 (McCracken and Wolf, 2019), Tarim as a transboundary river flows in China (area: 1048700 $km^2$, accounting for 95.5%), Kyrgyzstan (23900 $km^2$, 2.2%), disputed area between India and China and administered by China (22200 $km^2$, 2.0%), disputed area between India and China and administered by India (2000 $km^2$, 0.2%), Tajikistan (920 $km^2$, 0.1%) and Kazakhstan (110 $km^2$, <0.1%). In addition, some scholars also regarded the Tarim as transboundary river in their studies (De Stefano et al., 2017; Yan et al., 2019). Therefore, we think it is appropriate to discuss the Tarim River. We have added the explanation in the revised manuscript (Lines 310-312).

*Lines 310-312:*

*As well, there were 10 water political events (all cooperative) in the Tarim River Basin (a transboundary river basin among China, Kyrgyzstan, etc, according to TFDD), with water*

*quantity being the major theme.*

**References**

1. De Stefano, L., Petersen-Perlman, J. D., Sproles, E. A., Eynard, J., and Wolf, A. T.: Assessment of transboundary river basins for potential hydro-political tensions, Glob. Environ. Change-Human Policy, 45, 35-46, 2017.
2. McCracken, M. and Wolf, A. T.: Updating the register of international river basins of the world, Int. J. Water Resour. Dev., 35(5), 732-777, 2019.
3. Yan, J. B., Jia, S. F., Lv, A. F., and Zhu, W. B.: Water resources assessment of China's transboundary river basins using a machine learning approach, Water Resour. Res., 55(1), 632-655, 2019.

Ln640, figure 9. The size of line is hard to differentiate as the number of water conflictive events.

**-Author response:** Thank you for the kind comment. We have updated the figure and distinguished the number of water events by using the lines with different colors and widths (Lines 667-670).

*Lines 667-670:*

[Figure]

*Figure 10: Water conflictive and cooperative networks between Central Asian countries and other countries in the world: (a) Number of water conflictive events in 1951-1991 and (b) 1992-2018; (c) number of water cooperative events in 1951-1991 and (d) 1992-2018.*

Ln647, figure 10(b), the title of y-axis should be Number of water conflictive events? Check

it.

**-Author response:** Thank you for the instructive comment. Yes, the title of y-axis is "Number of water conflictive events". This figure (Figure 11(b) in the revised manuscript) is intended to show the intra-annual variations of water conflictive events. As is displayed in the Figure 11(b), a total of 53 water conflictive events were recorded in CA, nine of which occurred in January.

---

## Author Response (AR2)

We would like to thank two anonymous reviewers for their constructive suggestions. We have revised the manuscript based on the reviewer's recommendations. The followings are our point-by-point response and revisions to these comments.

**Response to Reviewer #1**

I am satisfied with the authors' responses to my comments, but some minor correction should be address:

**-Author response:** We thank Reviewer#1 for re-assessing our research and we highly appreciate the insightful suggestions and comments. We have further revised the manuscript based on the reviewer's recommendations. The followings are our point-by-point response and revisions to these comments.

I suggest the new figure 3 to be put in the supplementary material as it deviates the focus on the paper. Besides, same information has been shown in new figure 8.

**-Author response:** We thank the reviewer for the kind suggestion. We have moved the new Figure 3 to the supplementary material in the revised manuscript.

Please also clarify the why the number of water events differ in new figure 3 and 8.

**-Author response:** Thanks. We have checked the number of water events in these two figures. In fact, the number of water events in Figure 8 is the same as that in the TFDD events shown in Figure 3 for both conflictive and cooperative events prior to 2008 as these are derived from the TFDD database.